# Deep Edge Filter: Return of the Human-Crafted Layer in Deep Learning

**Dongkwan Lee**[*]  **Junhoo Lee**[*]  **Nojun Kwak**[†]

Seoul National University

{biancco,mrjunoo,nojunk}@snu.ac.kr

## Abstract

We introduce the Deep Edge Filter, a novel approach that applies high-pass filtering to deep neural network features to improve model generalizability. Our method is motivated by our hypothesis that neural networks encode task-relevant semantic information in high-frequency components while storing domain-specific biases in low-frequency components of deep features. By subtracting low-pass filtered outputs from original features, our approach isolates generalizable representations while preserving architectural integrity. Experimental results across diverse domains such as Vision, Text, 3D, and Audio demonstrate consistent performance improvements regardless of model architecture and data modality. Analysis reveals that our method induces feature sparsification and effectively isolates high-frequency components, providing empirical validation of our core hypothesis. The code is available at `https://github.com/dongkwani/DeepEdgeFilter`.

## 1 Introduction

Edge filters have long been a widely used classical technique in image processing to effectively capture relevant information, providing strong priors that are robust to various types of noise while effectively extracting semantic information. However, modern deep learning remains vulnerable to perturbation and domain shift [25, 23]. This is because the superficial low-level texture dependencies acquired by deep learning models during training further exacerbate their vulnerability to perturbations [8]. This vulnerability is evident in the fields of adversarial attack [10, 7] and domain adaptation [18, 42]. For example, even small adversarial perturbations, difficult for the human eye to detect, can cause significant confusion in models, or the performance of models can be greatly degraded simply because of a change from day to night.

Why have we lost knowledge of edge detection? Indeed, numerous past attempts have sought to integrate edge detection techniques into the deep learning domain [32, 48]. The core idea was to apply edge detection, or methodologies inspired by it, to input images. These processed images would then be fed into deep learning models, aiming to leverage the inherent robustness of edge detection against perturbations. However, such approaches failed to gain significant traction. This lack of success can be attributed to two primary factors. Firstly, while applying Edge Filters to images offers the benefit of robustness against perturbations, it conversely leads to the removal of fine-grained image details. This diminishes the informational content fed into deep learning models, limiting the ability to fully leverage the strengths of deep learning models, which excel at processing information-rich inputs. Secondly, classical edge detection was limited to the image domain. This inherent limitation makes it difficult to employ universally within the current deep learning landscape, which increasingly focuses on handling diverse data modalities, not only visual data but also a wide array of modalities including natural language, 3D scenes, and audio.

---

[*]Equal Contribution

[†]Corresponding Author

39th Conference on Neural Information Processing Systems (NeurIPS 2025).

In this paper, we generalize the concept of Edge Filters for deep features that can be applied directly to deep layers rather than the input layers, combining the benefits of traditional Edge Filters with deep learning to build models robust to perturbations and domain shifts. Our approach is motivated by our hypothesis that deep networks encode task-relevant semantic features at high frequencies and domain-specific biases at low frequencies. If this hypothesis is true, then generalizing Edge Filters (which essentially function as high-pass filters) should help isolate generalizable features. We contextualize this hypothesis based on domain adaptation research and validate it with a simple observation based on ResNet.

We define the Edge Filter applied to deep layers as the residual obtained by subtracting the result of low-pass filter (LPF) from the original. Here, we can utilize low-pass filters such as mean, median, and Gaussian kernels. When applying these LPFs to the deep features, the dimensionality of the kernels is naturally tailored to the specific deep learning architecture. Specifically, we applied 2-dimensional kernels to CNN-based architectures and 1-dimensional kernels to transformer-based architectures and MLPs.

We validate the universal effectiveness of Edge Filter across diverse modalities, tasks, and architectures. Our experiments span four fundamentally different modalities: Vision, Text, 3D, and Audio. For each modality, we selected tasks demanding strong generalization [16, 2, 19]: test-time adaptation [42, 14], sentiment analysis [41], few-shot neural radiance fields [24, 27], and audio classification [34]. Testing on both CNN-based [52] and Transformer-based [6] architectures confirmed architecture-agnostic applicability. We conducted ablation studies across filter configurations with different domains to demonstrate the consistent effectiveness of our approach. Furthermore, we found that applying the Edge Filter induces feature sparsification, in line with our theoretical sparse-coding framework. This was clearly demonstrated by the significant decrease in activation density following filter application. Moreover, we validate the need of high frequency proof-by-contradiction by showing that applying a low-pass filter impairs generalization ability.

We make the following contributions:

- We introduce Edge Filter, a filter built on human intuition, that can be applied to the feature of deep neural networks in a modality-agnostic manner to facilitate the extraction of generalizable features.

- We propose an Edge Filter for both CNN-based and ViT-based architectures, and empirically demonstrate that the filter enhances the performance of generalizability-crucial tasks across various modalities including image, text, 3D, and audio.

- We analyze the experimental results through the lens of layer sparsity and frequency decomposition, and provide extensive ablation studies on Edge Filters for deep features.

## 2 Related Works

### 2.1 Frequency Perspective in Deep Learning

Research on frequency analysis within deep learning has largely centered on image data and Convolutional Neural Networks (CNNs), offering valuable insights into how these models encode and process information. It has been observed that CNNs trained on ImageNet exhibit a strong bias towards texture rather than shape, a tendency distinct from human perception, and preferentially recognize textural cues in style-transferred images [8, 45]. Furthermore, Deep Neural Networks (DNNs) adhere to the 'Frequency Principle', learning low-frequency components earlier than high-frequency ones during training; this phenomenon is often interpreted as an implicit bias arising from the regularity of activation functions [49, 50].

Building on these understandings, various techniques have leveraged frequency concepts to enhance model performance and efficiency [48, 43]. In architectural innovations, the Fourier Convolutional Neural Network [11] has improved efficiency by performing convolutions in the frequency domain using Fast Fourier Transform, while similar efficiency benefits have been achieved through Discrete Cosine Transform operations [9]. From a training perspective, frequency domain data augmentation techniques have proven effective in preventing networks from overfitting to specific frequency components, thereby enhancing generalization capabilities [47, 40]. The capacity of frequency analysis to extract object location and boundary information has also led to successful applications in

object detection [26]. Despite these advancements, existing approaches have primarily focused on image modalities and rarely explored the direct application of frequency decomposition or filtering techniques to the internal learned representations within deep neural networks.

## 2.2 Activation Filtering and Sparsity

Filter Response Normalization (FRN) [37], when paired with a thresholded linear unit as its activation function, utilizes learnable thresholds to effectively filter activation values. This combination has demonstrated superior performance and versatility compared to conventional batch normalization or group normalization, exhibiting robustness particularly with small batch sizes. Deep Frequency Filtering [22] enhances domain generalization by directly filtering the hidden feature representations of a network within the frequency domain. It transforms feature maps into the frequency domain using FFT and then dynamically adjusts each frequency component with a learnable attention mask. Similarly, the Deep Edge-aware Filter [46] achieved model acceleration by implementing patch-wise processing and spatially varying filtering within an integrated deep learning framework.

Recent research on neural network training dynamics has contributed to understanding activation filtering. Studies on the 'edge of stability' [1, 4] show training under these conditions creates threshold neurons that activate selectively, promoting network sparsity and potentially improving generalization. ProSparse [39], designed for LLMs, combines ReLU functions with progressive sparsity regularization to achieve high activation sparsity, enhancing inference efficiency while minimizing performance loss. While these approaches prove effective in specific domains, they haven't explored general-purpose filtering layers applicable across diverse deep learning applications.

# 3 Method: Deep Edge Filters

## 3.1 Rethinking the Role of Edge Filters

We hypothesize that applying Edge Filters to deep features improves generalization by isolating semantic components and suppressing domain-specific features. This hypothesis is supported from a sparse coding perspective, with justification drawn from domain adaptation literature [42].

We define an Edge Filter as a high-pass operator constructed by subtracting a low-pass filtered version of $\mathbf{h}$, the original deep feature:

$$\mathcal{F}_{\text{edge}}(\mathbf{h}) = \mathbf{h} - \text{LPF}(\mathbf{h}), \quad (1)$$

where LPF denotes a low-pass filter such as a mean, median, or Gaussian kernel applied to $\mathbf{h}$.

The Edge Filter defined above is attached in series between the blocks of the base model, as shown in Fig. 1. When implementing the Edge Filter, parameters such as kernel size and other filter-specific parameters may be additionally required, depending on the type of filter.

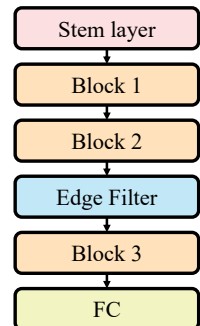

Figure 1: Example of a model architecture with Edge Filter attached. This figure illustrates the case when the Edge Filter is attached after Block 2 of the base model.

**Linear Probing Observation.** To validate our hypothesis, we conducted an observational study to examine the impact of low-pass filtering and high-pass edge filtering on linear probing performance. We loaded an ImageNet-pretrained ResNet18 model and attached filters to the output of its final block. While performing linear probing on the CIFAR-100 dataset, only the parameters of the fully connected (FC) classifier are updated, with the feature extractor frozen. For low-pass filtering (LPF), we employed a mean filter with a kernel size of 3. The model was trained for 10 epochs with a learning rate of 0.01.

The experimental results are presented in Tab. 1. We observed that applying LPF led to a decrease in performance compared to the unfiltered baseline, while application of an Edge Filter improved performance over the baseline. Given the nature of linear probing, where the feature extractor remains unchanged, the model must train the linear classifier using only the feature representations acquired

Table 1: Linear probing accuracy(%) on train and validation set of an ImageNet pre-trained ResNet18 on the CIFAR-100 dataset. Results compare three scenarios based on filtering applied to the deep features: Vanilla model (w/o Filter), Low-Pass Filter (LPF), and Edge Filter ($\mathcal{F}_{\text{edge}}$).

|           | w/o Filter | LPF  | $\mathcal{F}_{\text{edge}}$ |
|-----------|------------|------|------------------------------|
| train acc | 21.8       | 17.4 | **22.4**                     |
| val acc   | 21.0       | 17.4 | **21.4**                     |

during its ImageNet pretraining, rather than adapting them to the CIFAR-100 dataset. The performance observed with the Edge Filter suggests that it effectively filters out superficial information from the feature extractor, thereby better conveying task-relevant semantic signals to the classifier. This, in turn, indicates that deep learning models tend to encode domain-specific information in the low-frequency regions of deep features, while semantic information is predominantly encoded in the high-frequency regions.

**Decomposition of Deep Features.** Let $\mathbf{h} \in \mathbb{R}^d$ be a feature vector from a hidden layer of a deep network. We assume the feature can be additively decomposed as

$$\mathbf{h} = \mathbf{h}_{\text{sem}} + \mathbf{h}_{\text{dom}}, \tag{2}$$

where $\mathbf{h}_{\text{sem}}$ encodes generalizable, task-relevant semantic features, and $\mathbf{h}_{\text{dom}}$ represents domain-specific biases such as illumination, resolution, or background texture.

**Empirical Justification from Domain Adaptation.** Domain adaptation research offers strong empirical support for this decomposition. In particular, several studies have shown that simply aligning batch normalization (BN) statistics, the mean and variance of feature activations, between source and target domains leads to significant improvements in target accuracy [21, 42, 51, 44]. This suggests that domain-specific information is heavily embedded in the *first- and second-order statistics* of the features.

These statistics primarily capture the global, low-frequency structure of features, including texture, color tone, and overall feature scale. Hence, domain-specific components such as $\mathbf{h}_{\text{dom}}$ can be regarded as *low-frequency* in nature.

On the other hand, semantic components $\mathbf{h}_{\text{sem}}$, such as object boundaries, key phrases, or spatial discontinuities, tend to be spatially localized and structurally distinct. These elements are naturally more variable and correspond to *high-frequency* signal components.

**Sparse Coding through High-Pass Filtering.** Under our proposed feature decomposition and frequency assumptions, we have:

$$\text{LPF}(\mathbf{h}) \approx \mathbf{h}_{\text{dom}} \quad \Rightarrow \quad \mathcal{F}_{\text{edge}}(\mathbf{h}) \approx \mathbf{h}_{\text{sem}}. \tag{3}$$

This approach of refining features by frequency filtering resonates strongly with the principles of sparse coding. In sparse coding, the objective is to find a concise representation for features h by expressing them as a linear combination of a small subset of basis vectors from an overcomplete dictionary $\mathbf{D} \in \mathbb{R}^{d \times k}$:

$$\mathbf{h} \approx \mathbf{D}\boldsymbol{\alpha}, \quad \text{with} \quad \|\boldsymbol{\alpha}\|_0 \ll k. \tag{4}$$

The connection becomes clear when we consider the effect of our frequency filtering: by removing the low-frequency, domain-specific redundancies from $\mathbf{h}$ via edge filtering, we are essentially simplifying the signal that needs to be represented. This simplification naturally encourages greater sparsity in the corresponding coefficient vector $\boldsymbol{\alpha}$. The resulting feature representations, now focused on semantic content, are therefore more compact and possess enhanced generalizability across different domains.

Through the lens of domain adaptation, we argue that domain-specific features predominantly occupy the low-frequency spectrum of hidden activations, while task-relevant semantic content resides in the high-frequency components. The proposed edge filtering strategy leverages this structure to improve feature generality and downstream performance across diverse domains.

## 3.2 Deep Edge Filter

We propose a simple Deep Edge Filter applicable to deep features across all modalities. This filter functions as a high-pass operator, defined as the subtraction of the original feature and its LPF value, as specified in Eq. (1). For simplicity, we employ a mean filter as the default LPF throughout this paper unless otherwise stated. The Edge Filter is implemented channel-wise on each deep feature, with reflect padding applied to preserve input and output dimensions.

To match the dimensions of deep features, we use different types of Edge Filters for different architectures. For CNN-based architectures, we apply 2D Edge Filters because CNNs naturally process spatial relationships between neighboring pixels both vertically and horizontally. The 2D Edge Filter works by taking the $l$-th layer feature $\mathbf{h}_l \in \mathbb{R}^{H_l \times W_l \times C_l}$ and subtracting the LPF value across the spatial dimensions $(H_l, W_l)$ for each channel. For MLP and Transformer-based architectures, we apply 1D Edge Filters since these architectures don't inherently process spatial relationships. For these models, the 1D Edge Filter operates on the $l$-th layer feature $\mathbf{h}_l \in \mathbb{R}^{N_l \times C_l}$ by subtracting the 1D LPF value across the sequence length dimension $N_l$ for each channel.

The LPF component of the Deep Edge Filter is detached in the model training process to inhibit gradient backpropagation. This design choice reflects the non-learnable nature of the LPF, as our objective is to train the model exclusively on the high-pass filtered input. Furthermore, we implement the Edge Filter in only a single layer rather than introducing multiple filters within a model. This is because we have observed that using multiple filters in a model causes substantial information loss, making it difficult for the model to train properly.

# 4 Experiments

**Modality Selection.**  Deep learning models extract features differently depending on input data modality. Images and videos, derived from natural world observations, typically exhibit sparse distributions in their high-dimensional raw input space and contain inherent spatial patterns and visual attributes. In contrast, text data, generated according to human-defined linguistic rules, primarily encodes semantic relationships. Even when embedded into feature spaces of common dimensionality, features from distinct modalities retain unique, modality-specific properties due to their fundamentally different underlying distributions.

To test our claim rigorously, we conducted experiments on four modalities that have different characteristics. These modalities are Vision, which follows a nature-oriented distribution, Text, which follows human-defined rules and is defined in a discrete domain, 3D, which estimates spatial density, and Audio, which has temporal patterns and frequency characteristics. Our experiments demonstrate that our methodology, inspired by edge detection, can be generalized across deep features from diverse modalities and architectures.

**Overview.**  To evaluate the efficacy of our approach, we selected tasks across various modalities where generalization capability plays a crucial role. Through extensive experiments, we demonstrate consistent performance improvements, validating the hypothesis that Edge Filters effectively extract generalized features within deep neural networks. The following subsections correspond to experiments on selected tasks in each modality. Each subsection explains why the chosen task requires generalization capabilities and reports experimental results when applying Edge Filters. Implementation details are provided in Sec. A. All experiments were run using a single A6000 GPU.

## 4.1 Vision: Test-Time Adaptation

Test-Time Adaptation (TTA) [42] is a task that adapts models pre-trained on a source domain to a target domain without access to source data. A model's generalization capability is evaluated using datasets with significant domain shifts between source and target domains. Model adaptability improves when it captures generalizable and semantically meaningful patterns instead of overfitting to superficial or domain-specific cues. Therefore, better TTA performance indicates that the model has been effectively trained to extract general and semantic features during the learning process.

We compare the performance of vanilla models and models with attached Edge Filters, both pre-trained under identical conditions, when applying TTA algorithms. Model evaluation is conducted using the commonly used CIFAR-10C/100C and ImageNet200-C benchmarks [14] for TTA task.

Table 2: Test-time adaptation accuracy (%) results with and without filters as measured by the CIFAR-10C/100C and ImageNet200-C benchmarks with each TTA method applied.

| Method | CIFAR-10C | | | | CIFAR-100C | | | | ImageNet200-C | | | |
|---|---|---|---|---|---|---|---|---|---|---|---|---|
| | Source | Direct | NORM | TENT | Source | Direct | NORM | TENT | Source | Direct | NORM | TENT |
| WRN28-10 / ResNet18 | 91.9 | 49.6 | 73.8 | 74.6 | 69.8 | 26.2 | 46.6 | 48.0 | 58.8 | 2.0 | 23.9 | 21.6 |
| +$\mathcal{F}_{edge}$ | 90.8 | 57.7 (+8.0) | 75.3 (+1.5) | 75.8 (+1.2) | 65.4 | 36.4 (+10.2) | 49.7 (+3.1) | 50.5 (+2.5) | 55.5 | 2.1 (+0.1) | 25.8 (+1.9) | 23.2 (+1.6) |
| ViT-B/32 | 95.6 | 60.8 | - | 60.9 | 82.3 | 41.3 | - | 41.0 | 83.4 | 27.5 | - | 30.7 |
| +$\mathcal{F}_{edge}$ | 95.3 | 68.1 (+7.3) | - | 69.4 (+8.5) | 81.8 | 41.7 (+0.4) | - | 42.0 (+1.0) | 93.1 | 29.4 (+1.9) | - | 32.6 (+1.9) |

These benchmarks validate models trained on each clean dataset CIFAR-10/100 [17] and ImageNet200 [33], against corrupted versions of each dataset. The results are presented in Tab. 2. *Source* represents evaluation results on the original, uncorrupted validation sets, while *Direct* shows results when pretrained models are applied directly to the corrupted sets without adaptation. Results after applying TTA methods NORM [35] and TENT [42] are also included. For Vit, since there is no BatchNorm layer, the NORM algorithm could not be applied. We modified the TENT implementation to update parameters of LayerNorm layers instead of BatchNorm layers.

As shown in Tab. 2, applying the Edge Filter yields significant performance improvements in both backbones regardless of the TTA methodology used. Performance improvements ranged from 1.2%p to 8.5%p on CIFAR-10C, from 0.4%p to 10.2%p on CIFAR-100C, and from 0.1%p to 1.9%p on ImageNet200-C across different methods. Particularly noteworthy is that despite the reduction in performance on the *Source* dataset when applying the Edge Filter, performance on corrupted datasets improved. This suggests that while vanilla models tend to overfit to the training dataset during the training stage, the Edge Filter removes domain-specific information, preventing overfitting and encouraging the model to learn more general representations.

## 4.2 Language: Sentiment Analysis

General Language Understanding Evaluation (GLUE) benchmark is a standard evaluation tool that includes various natural language understanding (NLU) tasks [41]. We selected subtasks such as SST-2 (movie review sentiment analysis) [38], QQP (determining semantic equivalence of question pairs) [29], QNLI (determining if an answer is contained in a question) [31] for our experiments. A common characteristic of these language tasks is extracting specific discrete information (sentiment, semantic relationship, logical relationship) from vast high-dimensional text data (corpus). The crucial aspect in this process is effectively distinguishing necessary semantic information (signal) from unnecessary stylistic elements or additional information (noise). For example, in sentiment analysis, emotional expressions are signals while movie content details are noise. For language models to develop generalization capability, the ability to filter out such noise is important, and Edge Filter can assist by preserving semantic information in high-frequency components while removing domain-specific information in low-frequency components.

Experimental results are presented in Tab. 3. Performance improvements were observed in all GLUE subtasks when Edge Filter was applied. The most significant improvement (+1.49%p) occurred in the sentiment analysis task SST-2, which can be attributed to movie review data containing relatively more noise, including numerous movie-related additional information (actors, plot, scene descriptions, etc.) beyond sentiment expressions. Edge Filter effectively removes this noise, allowing the model to focus on information crucial for sentiment judgment.

Table 3: Classification accuracy (%) for GLUE benchmark tasks with and without $\mathcal{F}_{edge}$. Numbers in parentheses indicate performance improvement.

| | SST-2(Sentiment) | QQP(Paraphrase) | QNLI(Inference) | Avg |
|---|---|---|---|---|
| Transformer | 79.36 | 83.42 | 62.40 | 75.06 |
| +$\mathcal{F}_{edge}$ | 80.85 (+1.49) | 83.46 (+0.04) | 63.30 (+0.90) | 75.87 (+0.81) |

### 4.3 3D: Few Shot Neural Radiance Field

Neural Radiance Field (NeRF) [24] relies on generalization capabilities that leverage deep learning's inherent interpolation abilities to accurately represent complex 3D spaces using a simple MLP architecture. Particularly, its ability to model 3D scenes from a small number of 2D images and render from novel viewpoints demands generalization performance for unobserved viewpoints. These characteristics make it an appropriate modality for validating the efficacy of Edge Filter.

While 3D objects possess complex geometric structures, NeRF essentially focuses on volume boundary prediction, which consists of discontinuities at object surfaces and boundaries that exhibit high-frequency characteristics. Conversely, low-frequency features such as uniform densities within objects or empty spaces are relatively less significant. Edge Filter can enhance generalization capability by preserving these high-frequency components (signal) while removing low-frequency components (noise). Since NeRF processes 3D data through 1D ray synthesis, we applied a 1D Edge Filter to the layer immediately preceding density calculation in the coarse network.

For experiments, we used the widely-adopted Blender dataset[24], with results presented in Tab. 4. Performance improvements were observed in most scenes after applying Edge Filter, with an average enhancement of 5.2% for overall metrics. Most notably, the LPIPS metric decreased substantially to 11%. This supports our hypothesis because Edge Filter is designed to inject human prior. However, the scene *ficus* showed a 4.5% performance drop—likely because its naturally rich high-frequency textures, as shown in Fig. 2 rendered further enhancement by the Edge Filter redundant or even counterproductive.

Table 4: Few-shot NeRF rendering quality with 8-view inputs on various scenes. Numbers in parentheses indicate improvements.

| Scene | PSNR↑ NeRF | PSNR↑ $+\mathcal{F}_{edge}$ | SSIM↑ NeRF | SSIM↑ $+\mathcal{F}_{edge}$ | LPIPS↓ NeRF | LPIPS↓ $+\mathcal{F}_{edge}$ | MAE↓ NeRF | MAE↓ $+\mathcal{F}_{edge}$ |
|---|---|---|---|---|---|---|---|---|
| chair | 25.20 | **26.08 (+0.88)** | 0.911 | **0.917 (+0.006)** | 0.090 | **0.076 (-0.014)** | 0.018 | **0.016 (-0.002)** |
| drums | 17.12 | **17.31 (+0.19)** | 0.761 | **0.770 (+0.009)** | 0.242 | **0.223 (-0.019)** | 0.055 | **0.053 (-0.002)** |
| ficus | **22.33** | 22.12 (**-0.21**) | **0.889** | 0.881 (**-0.008**) | **0.075** | 0.084 (**+0.009**) | 0.024 | **0.025 (+0.001)** |
| lego | **24.58** | 24.53 (**-0.05**) | 0.894 | 0.894 (+0.000) | 0.074 | **0.071 (-0.003)** | 0.020 | 0.020 (+0.000) |
| mic | 27.64 | **28.17 (+0.53)** | 0.958 | **0.961 (+0.003)** | 0.049 | **0.041 (-0.008)** | 0.010 | 0.010 (+0.000) |
| ship | 20.80 | **22.19 (+1.39)** | 0.720 | **0.749 (+0.029)** | 0.224 | **0.175 (-0.049)** | 0.046 | **0.036 (-0.010)** |
| Avg | 22.95 | **23.39 (+0.44)** | 0.856 | **0.862 (+0.006)** | 0.126 | **0.112 (-0.014)** | 0.029 | **0.027 (-0.002)** |

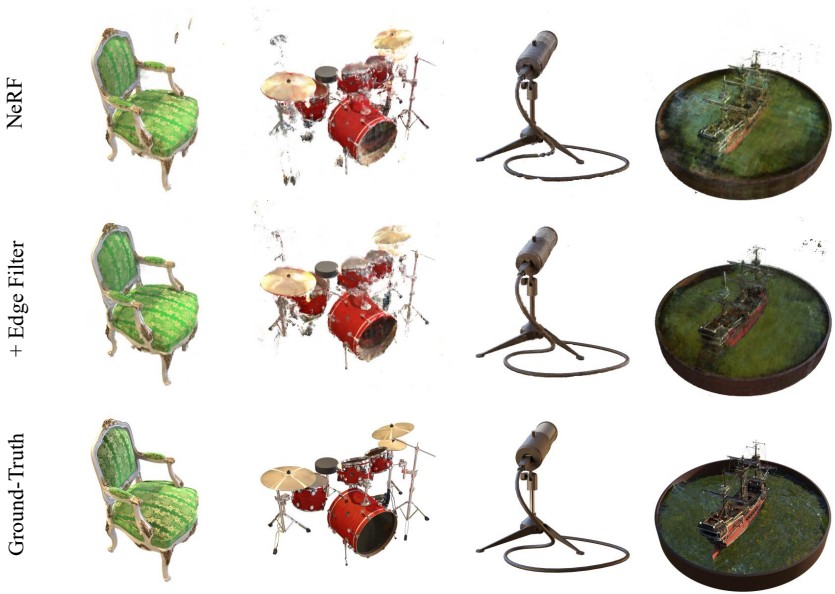

Figure 2: Qualitative analysis on NeRF. Overall, Edge filtering reduces floating artifacts.

Overall, Edge Filter was confirmed to effectively remove view-dependent noise while preserving essential information such as boundary surfaces in 3D representation, thereby improving generalization performance. As shown in Fig. 2, NeRF with Edge Filter demonstrates visually cleaner rendering results. In particular, 'floaters' (indistinct areas faintly floating around objects) commonly occurring in conventional NeRF are significantly reduced. This is because Edge Filter effectively removes superficial noise, such as incorrectly predicted volume densities, while preserving high-frequency signals like object boundaries. Overall, Edge Filter effectively eliminates view-dependent noise while preserving key information such as boundary surfaces in 3D representation, thus enhancing generalization performance.

### 4.4 Audio: Audio Classification

Audio classification tasks identify the type or source of sounds by extracting meaningful patterns from acoustic signals. In real-world environments, audio signals contain substantial noise and unnecessary background sounds, making it essential for classification models to effectively distinguish characteristic sounds (signal) from noise. Due to these properties, audio classification is another suitable modality for validating the effectiveness of Edge Filter. In our experiments, we used the UrbanSound8K dataset, which contains considerable noise acquired from urban environments. Experimental results showed that models with Edge Filter applied demonstrated performance improvements from 77.42% to 81.72% compared to baseline models. This improvement is attributed to Edge Filter's ability to emphasize characteristic sound patterns while effectively removing background noise, thereby enhancing classification performance.

## 5 Analysis

To investigate the impact of the Deep Edge Filter on model training, we measured the density of layer outputs throughout the training process. Fig. 3a and Fig. 3b demonstrate the output density of each WRN-28-10 layer by training epoch during the pretraining stage on the CIFAR-10 dataset as described in Section 4.1. Density values represent averages across the entire training dataset for each epoch. For the *conv* layer and *block1*, which are positioned before the Edge Filter, minimal density differences were observed. In contrast, *block2* and *block3*, which are positioned after the Edge Filter, show significantly decreased output density when the Edge Filter is present. This effect is most pronounced in *block2*, which is located right after the filter. This dramatic reduction in feature density following filter application experimentally validates the sparse coding of features resulting from high-pass filtering, as formulated in Eq. (4).

Additionally, we performed FFT analysis to examine how the Deep Edge Filter influences the frequency domain characteristics of deep features. Fig. 3c demonstrates the average FFT results for the filter's input and output on the CIFAR-10 validation set after completing the training shown in Fig. 3b. The graph shows the x-axis cross-section at the center of the 2D FFT, averaged across channels. The FFT results demonstrate that the amplitude in the low-frequency region of deep features

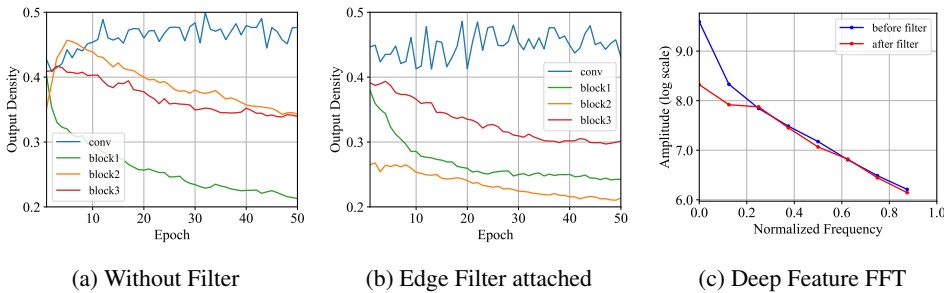

|(a) Without Filter|(b) Edge Filter attached|(c) Deep Feature FFT|

Figure 3: Statistical analysis of the impact of Edge Filter on deep features. Fig. 3a and Fig. 3b show the sparsity of output features for each block when training a vanilla model and a WRN-28-10 model with Edge Filter attached to block1 on the CIFAR-10 dataset, respectively. Fig. 3c represents the FFT amplitude spectra of both inputs and outputs from the Edge Filter when evaluated on the validation set after the training in Fig. 3b is completed.

decreases after passing through the Edge Filter. This confirms that the Edge Filter, designed as a high-pass operator through LPF subtraction, functions as intended.

## 5.1 Ablation studies

Fig. 4 illustrates ablation studies conducted to examine the effects of Edge Filter's attachment position and kernel size. Performance gains compared to the baselines are represented as heatmaps. Layer 0 on the vertical axis denotes the stem network (*conv* layer) of CNN-based models. Figures 4a and 4b present results from applying various filter configurations to the WRN-28-10 and ViT-B/32 models, respectively, on the CIFAR-10C benchmark using the *Direct* method. The WRN model shows a general trend of performance improvement with the Edge Filter application, achieving a maximum improvement of 9.6%p. For the ViT model, performance improvements were pronounced when filters were applied to later layers, while applying filters to earlier layers tended to decrease performance. Figure 4c shows results from applying filters with various configurations on SST-2, a language domain task. Performance either remained unchanged or improved regardless of the Edge Filter's position and kernel size, with a maximum improvement of 3.6%p. In the audio domain, as shown in Figure 4d, performance fluctuated with variations ranging from -2.0%p to +3.6%p without exhibiting any clear trends. The statistical significance test results for the baseline of each task are shown in Sec. B.

Table 5: Ablation study on vision domain (TTA) and language domain (SST-2) for filter types.

| Modality | Vision | | Language |
|---|---|---|---|
| Benchmark | CIFAR-10C (Direct) | | SST-2 |
| Backbone | WRN | ViT | BERT |
| Baseline | 49.64 | 60.84 | 79.36 |
| +Mean $\mathcal{F}_{edge}$ | 57.65 (**+8.01**) | 68.09 (**+7.25**) | 80.85 (**+1.49**) |
| +Median $\mathcal{F}_{edge}$ | 59.69 (**+10.05**) | 67.28 (**+6.44**) | 81.60 (**+2.24**) |
| +Gaussian $\mathcal{F}_{edge}$ | 57.66 (**+8.02**) | 56.37 (**-4.47**) | 80.58 (**+1.12**) |
| +Mean LPF | 39.69 (**-9.95**) | 47.96 (**-26.68**) | 57.56 (**-3.28**) |

Tab. 5 presents an ablation study examining how model performance changes when different filters replace the mean filter in the LPF component of the Edge Filter. We compared mean, median, and Gaussian filters as LPFs, and also tested applying the mean filter (LPF) directly to deep features instead of using the Edge Filter. Our experiments utilized WRN and ViT backbones for vision tasks and the BERT backbone for language tasks, with filter configurations matching those in 4 and the Gaussian filter's sigma set to 1.0. Results demonstrate that performance improvements remained consistent across different LPF types within the Edge Filter, indicating that various LPFs can be effectively incorporated into the Deep Edge Filter design. We observed a performance decline when using the Gaussian filter with ViT, likely due to using a default sigma value rather than optimizing it. When applying LPF directly to deep features, significant performance degradation occurred across all tasks, confirming that low-frequency components of deep features tend to encode domain-specific information, causing performance degradation in tasks requiring generalizability.

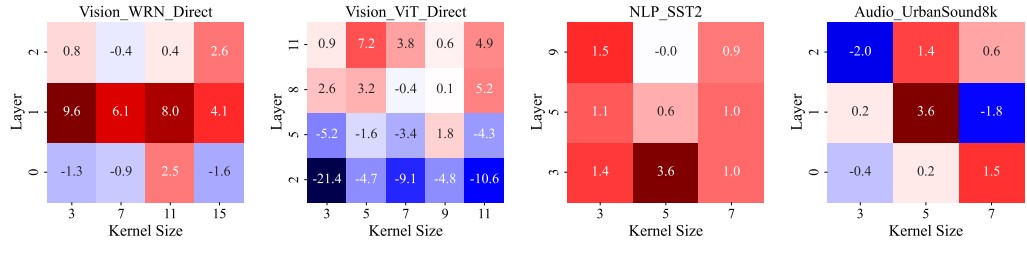

(a) Vision, TTA on WRN    (b) Vision, TTA on ViT    (c) Language, SST2    (d) Audio, UrbanSound8k

Figure 4: Heatmap of performance gain (%p) compared to the cases without Edge Filter, depending on the position and the kernel size of the Edge Filter. CIFAR-10C TTA benchmark is used in the vision domain, SST-2 in the language domain, and UrbanSound8k classification in the audio domain.

# 6 Conclusion

In this paper, we show that Edge Filters can be applied directly to deep features in deep learning models to extract more generalized features from input data. This is based on the observation that deep learning models encode task-relevant semantic information in the high-frequency region, while domain-specific information in the low frequency region. Since domain-specific information often acts as a superficial feature in semantic inference tasks, filtering out the low frequencies can improve the model's generalization ability. Experiments on various domains such as vision, language, 3D, and audio, validate the effectiveness of the proposed Deep Edge Filter and show that it is broadly applicable regardless of the data modality.

**Limitations and Future Works.** Because our protocol requires retraining models from scratch after attaching the Edge Filter, we were unable to conduct more extensive experiments due to computational cost constraints. A limitation of this work is that we could not validate the methodology's effectiveness on state-of-the-art models or across a wider variety of tasks. This limitation is particularly pronounced in the language domain, where experiments with LLMs were infeasible due to their high computational demands. Future work applying our approach to LLMs would provide valuable extensions to our claims from an application perspective. Additionally, given the domain-agnostic characteristics of the Deep Edge Filter, exploring its application to multimodal models represents another promising direction for future research.

## Acknowledgments and Disclosure of Funding

This work was supported by NRF grant (2021R1A2C3006659) and IITP grants (RS-2022-II220953, RS-2021-II211343, RS-2025-25442338), all funded by MSIT of the Korean Government.

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

# A Implementation Details

## A.1 Vision Domain

Following the convention of the benchmarks, we used two Convolution-based models and one Transformer-based model: the CNN-structured WideResNet28-10 [52] for CIFAR datasets, ResNet18 [13] for ImageNet200 dataset, and the Transformer-structured ViT-B/32 [6] for all datasets. Learning rates of 1e-3 were applied to the convolution-based models, and 5e-5 was applied to ViT. Convolution-based models were trained from scratch, while ViT model was trained from the ImageNet1k pre-trained weights. We conducted training for 50 epochs for CIFAR10/100, while 5 epochs for ImageNet200, with a batch size of 128. The severity level of dataset corruptions was set to 5. For WRN28-10, 2D Edge Filter with kernel size 11 was applied after block 1 out of three blocks, while for ResNet18, kernel size 11 was applied after block 2 out of four blocks. For ViT-B/32, 1D Edge Filter with kernel size 5 for the CIFAR datasets and kernel size 3 for the ImageNet200 dataset after block index 11 out of the block indices ranging from 0 to 11.

## A.2 Language Domain

We used the vanilla 12-layer transformer, which is the standard BERT architecture [5], training from scratch, and applied a 1D Edge Filter with kernel size 3 after block 9. For training, we used AdamW optimizer, with a batch size of 512, a learning rate of 5e-5, and a weight decay of 1e-2 for 40 epochs.

## A.3 3D Domain

Experiments were conducted in the 8-view few-shot [27] setting using various 3D objects (ship, mic, lego, ficus, drums, chair) from the Blender dataset [24]. We followed the same optimization settings as the default NeRF implementation on the Blender dataset, and the kernel size is set to 5. Training was performed for 500 pixel epochs, and we used PSNR, SSIM, LPIPS [53], and MAE metrics for performance evaluation to measure different aspects of rendering quality. We applied a 1D Edge Filter with kernel size 5 to the coarse network prior to computing the density $\sigma$.

## A.4 Audio Domain

We used the UrbanSound8K [34] dataset, which includes 10 urban sound classes (e.g., car horn, children playing, dog barking). The model architecture consists of three convolutional blocks, and we add 2D Edge Filter with a kernel size of 7 after the third layer. The model was trained for 20 epochs using the Adam optimizer with a learning rate of 0.001.

# B Statistical Analysis

We conducted a replication study using multiple experimental runs with different random seeds to assess the statistical significance of the results presented in Fig. 4. Given the observed performance improvements over the baseline condition shown in Fig. 4, we calculated the mean and standard error of baseline performance metrics across replicate experiments to determine whether the effects of the Edge Filter method achieve statistical significance. Tab. 6 presents the means and standard deviations computed from five independent experimental runs using different random seeds for each modality under investigation.

Table 6: Statistical analysis on baseline performances (%) across modalities

| Domain | Vision | | Language | Audio |
|---|---|---|---|---|
| Benchmark | CIFAR10C (Direct) | | SST-2 | UrbanSound8k |
| Backbone | WRN | ViT | BERT | CNN |
| Mean | 49.37 | 61.84 | 79.89 | 76.99 |
| SD | 2.46 | 2.47 | 0.52 | 0.98 |

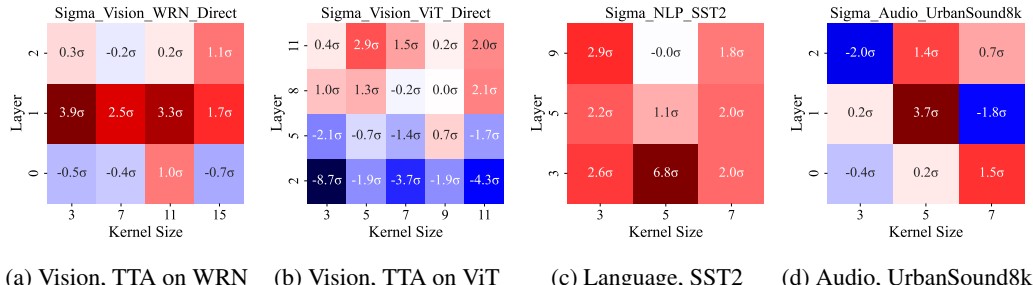

(a) Vision, TTA on WRN   (b) Vision, TTA on ViT   (c) Language, SST2   (d) Audio, UrbanSound8k

Figure 5: Heatmap of performance gains as **multiples of standard deviation ($\sigma$)**, represents the statistical significance of Fig. 4.

To evaluate the statistical significance of the performance improvements, we demonstrate the normalized performance analysis by plotting the performance gains expressed in standard deviations ($\sigma$) relative to the baseline standard deviation for each experimental condition, as presented in Fig. 5. This analysis shows that the performance gains across multiple modalities and backbone architectures exceed two standard deviations ($2\sigma$) from the baseline mean, thereby demonstrating statistically significant improvements attributable to the Edge Filter.

## C   T-SNE Visualizaiton

We performed t-SNE visualization on vision domain data to visually confirm the effect of the Edge Filter on deep learning models. Figures Fig. 6 and Fig. 7 show the t-SNE visualization results for the CIFAR-10 validation set extracted from a model trained on the CIFAR-10 dataset and for the features of CIFAR10-C, which is the CIFAR-10 dataset with added corruption, respectively. The WRN and ViT models, along with the attached mean filter, are identical to those used in Sec. 4.1.

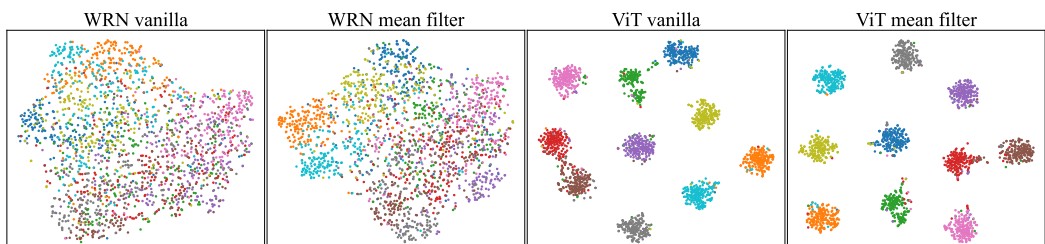

Figure 6: Comparison of t-SNE visualizations for vanilla WRN and ViT backbones with and without Mean Filters applied to CIFAR10 validation set.

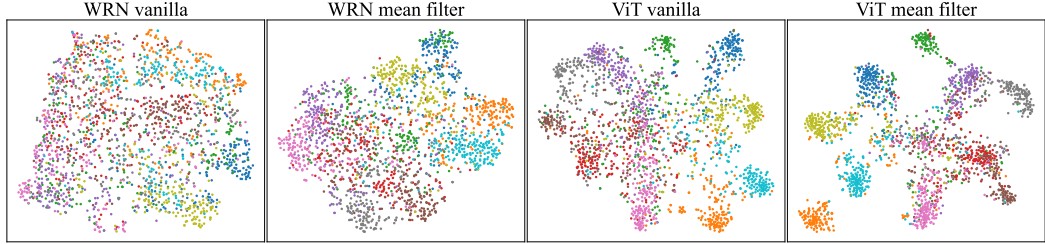

Figure 7: Comparison of t-SNE visualizations for vanilla WRN and ViT backbones with and without Mean Filters applied to **corrupted images** of CIFAR10 validation set, CIFAR10-C.

Visualization results on the clean validation set, as seen in Fig. 6, showed no distinct difference in feature distributions between the vanilla model and the model with a mean filter attached, for both the WRN and ViT backbones. Meanwhile, visualization results for corrupted images, as seen

in Fig. 7, show that the model generally struggles to distinguish classes well compared to clean images, resulting in blurred class clusters. Comparing the model with the mean filter attached to the vanilla model here reveals that, regardless of the backbone, the model with the mean filter exhibits relatively clear boundaries between classes. Notably, the difference in distribution between models with and without the mean filter is even more pronounced on the ViT backbone. These visual results align with the numerical findings in Tab. 2, confirming the role of the Edge Filter in enhancing model robustness, improving performance over the vanilla model on corrupted images.

## D    Full Experiment Result for TTA

Tab. 7 and Tab. 8 break down the results of Tab. 2 for 15 different corruption types [14]. They show that Edge Filter improves performance for the vast majority of corruptions.

Table 7: Classification accuracy (%) for all corruptions in CIFAR10-C

| Method | Backbone | Noise | | | Blur | | | | Weather | | | | Digital | | | | Avg |
|---|---|---|---|---|---|---|---|---|---|---|---|---|---|---|---|---|---|
| | | Gau | Sht | Imp | Def | Gls | Mtn | Zm | Snw | Frs | Fog | Brt | Cnt | Els | Px | Jpg | |
| Direct | WRN28-10 | 17.9 | 22.3 | 22.4 | 47.8 | 40.2 | 52.1 | 55.2 | 69.1 | 60.0 | 68.1 | 86.7 | 33.8 | 60.7 | 44.1 | 64.2 | 49.6 |
| | $+\mathcal{F}_{edge}$ | 42.2 | 48.7 | 36.4 | 56.3 | 50.5 | 64.1 | 63.3 | 70.6 | 65.7 | 67.8 | 86.2 | 33.1 | 68.1 | 43.3 | 68.7 | 57.7 |
| NORM | WRN28-10 | 64.9 | 67.4 | 56.5 | 82.6 | 58.8 | 80.0 | 82.4 | 76.8 | 76.1 | 80.7 | 87.6 | 80.2 | 70.6 | 72.9 | 69.0 | 73.8 |
| | $+\mathcal{F}_{edge}$ | 68.4 | 70.5 | 59.4 | 82.7 | 60.8 | 80.5 | 83.2 | 76.5 | 77.7 | 80.7 | 87.4 | 83.2 | 72.6 | 74.8 | 71.1 | 75.3 |
| TENT | WRN28-10 | 65.7 | 68.8 | 58.1 | 83.2 | 60.0 | 80.4 | 83.4 | 77.7 | 76.4 | 81.6 | 87.9 | 81.0 | 71.5 | 74.2 | 69.9 | 74.6 |
| | $+\mathcal{F}_{edge}$ | 69.2 | 71.4 | 60.5 | 83.1 | 61.6 | 80.8 | 83.7 | 76.8 | 78.3 | 81.1 | 87.7 | 83.6 | 72.9 | 75.4 | 71.5 | 75.8 |
| Direct | ViT-B/32 | 61.8 | 65.7 | 37.5 | 51.9 | 65.2 | 53.4 | 59.0 | 77.5 | 76.3 | 39.7 | 85.2 | 15.4 | 69.1 | 80.0 | 75.0 | 60.8 |
| | $+\mathcal{F}_{edge}$ | 67.5 | 71.8 | 46.4 | 70.2 | 69.8 | 66.9 | 72.2 | 80.1 | 79.9 | 55.5 | 85.0 | 20.7 | 75.9 | 80.7 | 79.0 | 68.1 |
| TENT | ViT-B/32 | 62.9 | 66.4 | 31.2 | 51.6 | 66.6 | 53.9 | 60.5 | 78.5 | 78.0 | 37.4 | 86.0 | 13.9 | 70.5 | 80.8 | 75.6 | 60.9 |
| | $+\mathcal{F}_{edge}$ | 69.2 | 73.3 | 46.2 | 73.4 | 70.3 | 69.2 | 73.9 | 80.5 | 81.3 | 56.1 | 85.5 | 23.8 | 76.7 | 81.8 | 79.4 | 69.4 |

Table 8: Classification accuracy (%) for all corruptions in CIFAR100-C

| Method | Backbone | Noise | | | Blur | | | | Weather | | | | Digital | | | | Avg |
|---|---|---|---|---|---|---|---|---|---|---|---|---|---|---|---|---|---|
| | | Gau | Sht | Imp | Def | Gls | Mtn | Zm | Snw | Frs | Fog | Brt | Cnt | Els | Px | Jpg | |
| Direct | WRN28-10 | 10.2 | 11.7 | 6.5 | 24.0 | 18.8 | 29.8 | 29.3 | 36.4 | 26.5 | 34.4 | 57.3 | 16.7 | 41.4 | 16.8 | 33.3 | 26.2 |
| | $+\mathcal{F}_{edge}$ | 25.3 | 27.4 | 14.2 | 33.6 | 39.7 | 35.7 | 39.2 | 42.6 | 42.9 | 32.0 | 52.1 | 14.5 | 46.4 | 50.9 | 48.9 | 36.4 |
| NORM | WRN28-10 | 33.8 | 34.3 | 26.8 | 57.0 | 37.2 | 53.5 | 57.3 | 46.5 | 47.4 | 50.6 | 63.4 | 55.0 | 46.7 | 50.0 | 38.7 | 46.6 |
| | $+\mathcal{F}_{edge}$ | 42.3 | 43.4 | 33.2 | 58.1 | 44.5 | 54.7 | 58.3 | 49.3 | 50.9 | 47.2 | 60.7 | 45.1 | 50.5 | 56.3 | 50.8 | 49.7 |
| TENT | WRN28-10 | 36.1 | 36.4 | 28.6 | 58.1 | 38.4 | 54.4 | 59.4 | 47.3 | 48.5 | 51.6 | 63.7 | 56.8 | 48.1 | 52.0 | 40.1 | 48.0 |
| | $+\mathcal{F}_{edge}$ | 43.9 | 44.4 | 34.2 | 58.9 | 45.1 | 54.9 | 58.5 | 50.2 | 52.0 | 48.1 | 60.8 | 46.1 | 51.4 | 57.4 | 51.1 | 50.5 |
| Direct | ViT-B/32 | 37.5 | 41.1 | 21.0 | 37.7 | 35.4 | 37.4 | 43.9 | 55.7 | 53.7 | 27.6 | 67.4 | 8.0 | 46.6 | 57.3 | 48.6 | 41.3 |
| | $+\mathcal{F}_{edge}$ | 34.2 | 38.5 | 17.8 | 42.3 | 36.3 | 41.2 | 48.1 | 53.9 | 55.2 | 30.9 | 64.4 | 9.9 | 48.3 | 51.7 | 52.3 | 41.7 |
| TENT | ViT-B/32 | 40.0 | 42.5 | 21.5 | 35.5 | 34.6 | 35.3 | 43.5 | 57.4 | 54.5 | 22.8 | 68.4 | 5.4 | 46.5 | 58.9 | 48.8 | 41.0 |
| | $+\mathcal{F}_{edge}$ | 34.8 | 39.1 | 17.4 | 43.1 | 35.9 | 41.9 | 49.7 | 54.8 | 56.3 | 28.2 | 65.7 | 8.2 | 48.8 | 53.4 | 52.7 | 42.0 |

## E    Expansion to Foundational Models

We validate whether our hypothesis remains effective in larger foundation models. To verify the effect of Edge Filter, we adopt a few-shot prototype matching evaluation protocol [12, 3] that measures the quality of the encoder's feature representations. Specifically, we construct prototypes by averaging the features of samples belonging to each class, and measure classification performance based on these prototypes.

We select foundation model ViT-L/14 OpenCLIP [15, 30] trained on the LAION2B dataset [36]. This model has learned robust representations across diverse visual domains through large-scale web-scale data training, making it well-suited for evaluating domain generalization capabilities.

We conduct cross-domain generalization experiments to evaluate whether the Edge Filter is also effective for larger models. This task measures the performance when prototypes extracted from one domain are applied to samples from another domain. We conduct Photo-to-Sketch, Sketch-to-Photo, Real-to-Sketch, and Sketch-to-Real transfer scenarios on the PACS [20] and DomainNet [28] datasets. As a result, we observed that applying Edge Filter also improves performance for larger foundation model, as shown in Tab. 9.

Table 9: Cross-domain few-shot classification accuracy (%) on PACS and DomainNet datasets. Numbers in parentheses indicate improvements over the baseline.

| | PACS: Photo to Sketch | | PACS: Sketch to Photo | | DomainNet: Real to Sketch | | DomainNet: Sketch to Real | |
| --- | --- | --- | --- | --- | --- | --- | --- | --- |
| Method | Baseline | $+\mathcal{F}_{edge}$ | Baseline | $+\mathcal{F}_{edge}$ | Baseline | $+\mathcal{F}_{edge}$ | Baseline | $+\mathcal{F}_{edge}$ |
| Accuracy | 59.07 | **63.22** (+4.15) | 54.31 | **60.12** (+5.81) | 24.79 | **25.00** (+0.21) | 39.23 | **39.53** (+0.30) |

## F  Effect of increased computation

To demonstrate that the performance improvement from the Edge Filter is not merely an effect of increased computation, we additionally conducted experiments comparing its performance with a model where the Edge Filter was replaced by a Convolution layer with the same computational load. Since it is widely known that performance tends to improve as the computational load of deep learning models increases, comparing these two cases with identical computational loads allows us to isolate the pure effect of the Edge Filter, excluding the impact of increased computation. The table below Tab. 10 shows the results of replacing the Edge Filter with a trainable 2D Conv layer of the same kernel size in the CIFAR10-C WRN28-10 experiment from Tab. 2. All other experimental conditions are identical to those used with the original Edge Filter.

Table 10: Comparison of CIFAR10-C TTA performance when replacing the Edge Filter with a trainable 2D Conv layer with equivalent computational load

| | Source | Direct | NORM | TENT |
| --- | --- | --- | --- | --- |
| No filter | **91.9** | 49.6 | 73.8 | 74.6 |
| Conv layer | 90.6 | 46.9 | 71.3 | 72.3 |
| Edge filter | 90.8 | **57.7** | **75.3** | **75.8** |

As seen in Tab. 10, replacing the Edge Filter with a trainable Convolution layer of equivalent computation load actually resulted in decreased performance on corrupted images. This demonstrates that the Edge Filter's effectiveness holds true even when accounting for the increased computational load introduced by its addition.

