# OpenReview forum: "Deep Edge Filter: Return of the Human-Crafted Layer in Deep Learning"
_NeurIPS.cc/2025/Conference — NeurIPS 2025 poster_

### Official Review · Reviewer_kGMU · 2025-06-30

**Clarity:** 2
**Significance:** 3
**Originality:** 3
**Rating:** 5
**Confidence:** 4

**Summary:**

This paper introduces the Deep Edge Filter, a high-pass filter for deep features, to improve model generalizability. Its key innovations include: (i) a modality-agnostic Edge filter that can facilitate the extraction of generalizable features; (ii) a simple but effective hand-crafted edge filter operator for both CNN and ViT architectures, demonstrating good cross-task generalizability across various modalities including image, text, 3D and audio. Ablation studies demonstrate the effectiveness w.r.t. layer sparsity and frequency.

**Questions:**

1.	Does the position of the mean filter applied significantly influence the results? Have the authors tried to apply the filter to alternative layers? In addition, does multiple edge filters for multiple layers help?
2.	Whether the edge filter could work by training from scratch, or is its efficacy is limited to pre-trained models? Additional experimental results addressing this point may strengthen the application scenario of the proposition.

**Ethical Concerns:**

["NO or VERY MINOR ethics concerns only"]

**Final Justification:**

The authors have engaged actively to provide additional results and persuaded other reviewers. I believe this is an interesting work that deserves the attention from the community. My final recommendation is Accept.

**Limitations:**

yes

**Quality:**

3

**Strengths And Weaknesses:**

Major Strengths
1. 	This worke finds an interesting pattern within deep features that a simple use of hand-crafted mean filter can decouple the semantic components from the domain-specific components within deep features. As far as I can tell, this finding, while being empirical, is novel and non-trivial.
2. The deep edge operator is validated across a diverse range of multi-modal data, substantiating its universal applicability to models and data of differing modalities.
3. The proposed fix is simple yet incredibly effective, exhibiting significant potential to extend to a wide range of tasks.

Major Weaknesses
1.	The core hypothesis, h=h_dom+h_sem, relies predominantly on empirical evidence from prior literature without providing direct evidence; the paper only presents preliminary experiments indicating improved model performance upon removal of low-pass components.
2.  The claim that “we generalize the concept of Edge Filters for deep features that can be applied directly to deep layers rather than the input layers” may be overclaimed, as learned convolutional filters are also a form of edge filters, potentially misleading readers regarding the contribution of the work.
3.  The formulation may lack robustness; for instance, in “Sparse Coding through High-Pass Filtering”, the authors may fail to elucidate the nature of high-pass filtering under sparse coding, while merely presenting the high-pass filtering equation (3) and the sparse coding equation (4) alone.

Minor Weaknesses
Although the study encompasses multiple modalities, the experimental comparisons within individual modalities are limited by a few methods and the absence of SOTA approaches.

---

> ### Author Rebuttal · Authors · 2025-07-31
>
> We appreciate the insightful feedback from the reviewer commenting on our method’s novel and non-trivial finding, experiments on broad modalities, and potential for wide applicability.
>
> **W1 Core hypothesis:** The reviewer seems to be asking about equation 3, which states that LPF(h) contains domain-specific information, while HPF(h) contains mostly domain-general information. This equation represents our hypothesis, and as the reviewer pointed out, our paper does not provide direct evidence (or proof), nor do we believe it is possible to do so. This is because our hypothesis deals with the features of deep networks, which cannot be handled in the same way as shallow networks, where analytical solutions exist. Instead of a mathematical proof, we conducted experiments related to generalizability across a wide range of modalities to empirically validate our claims. Furthermore, in this rebuttal phase, we additionally demonstrated that our hypothesis holds scale-invariant on larger datasets (ImageNet, PACS, DomainNet) and larger model (ViT-L-14 OpenCLIP). We hope that these various experiments supporting Eq. 3 will help convince you.
>
> **W2 Conv layer as edge filter:** I agree with the reviewer's comment that learnable conv layers can sometimes act like edge filters. However, just because convolutional filters can play the role of edge filters does not mean that convolutional filters always become edge filters. Rather, learned conv layers generally do not have a clear role in filtering specific frequencies like LPF or HPF, as their weights change depending on the learning process. The filter proposed in this paper is not learnable and was designed with the explicit intention of being an HPF, contrasting with the conv layer. For a clearer explanation, I have included the experimental results from the CIFAR-10 WRN experiment in Table 2, where the edge filter was replaced with a 3x3 learnable conv layer.
>
> |             | Source | Direct | NORM |  TENT |
> |-------------|:------:|:------:|:----:|:-----:|
> | No filter   |  **91.9**  |  49.6  | 73.8 |  74.6 |
> | Conv layer  |  90.62 |  46.92 | 71.3 | 72.32 |
> | Edge filter |  90.8  |  **57.7**  | **75.3** |  **75.8** |
>
> The results of applying the 3x3 conv layer showed lower performance than when applying the edge filter in all cases; Validation (source), corrupted dataset (Direct, NORM, TENT). Showing that the performance of extracting generalizable features was still insufficient due to overfitting on the train data.
>
> **W3 Formulation of Eq.3 and Eq.4:** This formula is not presented for robustness or theoretical rigor, but rather conceptually to explain our perspective to readers. Our perspective is that, when addressing issues related to generalization, certain features are superficial, and from this perspective, the problem naturally transitions to sparse coding. In the generalization problem, the truly important features are sparse, and our focus is on selecting these semantic features. From this conceptual algorithmic perspective, the transition from Equation 3 to Equation 4 is natural.
>
> **minorW Few methods and absence of SOTA approaches:** Since this paper covers a wide range of modalities, please understand that the types of tasks performed for each modality are limited. However, during this rebuttal process, we additionally performed SSDG tasks and comparisons with SOTA methods, as well as zero-shot classification (prototype matching) using the Open-CLIP model, and have attached the results of these experiments. We hope that these experiments will help address the reviewers' concerns.
>
> We applied our proposed method to a semi-supervised DG task, one of the most challenging tasks in the DG field. We used UPCSC[1], the SOTA method presented at CVPR 2025 as the baseline method. We kept the model architecture and training procedure unchanged and conducted experiments using one labeled sample per class from the PACS dataset. We applied the mean HPF of kernel size 11 on layer 2 of ResNet18. The results of the SSDG experiment are as follows.
>
> |                       |   A  |   C  |   P  |   S  |  avg |
> |---------------------|:----:|:----:|:----:|:----:|:----:|
> | UPCSC | 34.3 | 20.2 | 17.2 | 22.9 | 23.7 |
> |+ Edge Filter     | **41.6** | **43.9** | **21.1** | **47.9** | **38.6** |
>
> We also conducted prototype matching zero-shot classification experiments based on the ViT-L-14 OpenCLIP model trained with the Laion2B dataset. We used the PACS and DomainNet datasets as benchmarks. Specifically, we randomly selected one image per class from the source domain as the class prototype and compared the zero-shot classification accuracy of the shifted target domain images using this prototype. For PACS, we selected the Photo-Sketch domain, and for DomainNet, we selected the Real-Sketch domain. The performance of the baseline case, where the embedding vector used for prototype matching was used as-is, and the case where the Edge Filter was applied, is shown in the table below.
>
> |             |                |      PACS      |      | \| |               |   DomainNet   |      |
> |:-----------:|:--------------:|:--------------:|:----:|----|:-------------:|:-------------:|:----:|
> |             | Photo > Sketch | Sketch > Photo |  avg | \| | Real > Sketch | Sketch > Real |  avg |
> |   baseline  |      59.1      |      54.3      | 56.7 | \| |      24.8     |      39.2     | 32.0 |
> | Edge Filter |  **63.2**    |  **60.1**   | **61.7** | \| |   **25.0**  |  **39.5**  | **32.3** |
>
> **Q1 Filter position:** The position of the filter affects performance, and the degree of difference varies depending on the modality. As shown in Fig. 3, we conducted an extensive ablation study on filter position and kernel size in the vision, language, and audio modalities. In addition, when applying multiple edge filters, as described in lines 184–187, information loss becomes severe, resulting in poor performance.
>
> **Q2 Initial weight:** We didn't get to talk about this part in detail, so thank you for pointing it out. Except for the ViT backbone of the vision modality, all of our experiments were trained from scratch. We based our approach on training from scratch, but in the case of ViT, meaningful model training was impossible using only the given image dataset, so we had no choice but to use a pretrained model. In other words, existing experimental results have already demonstrated the applicability of our methodology in cases where both training from scratch and pretrained models were applied. We will clarify this point in the implementation details to avoid any confusion.
>
> **References:** [1] Lee, Dongkwan, et al. "Unlocking the Potential of Unlabeled Data in Semi-Supervised Domain Generalization." Proceedings of the Computer Vision and Pattern Recognition Conference. 2025.

---

> > ### Author Response · Authors · 2025-08-04
> >
> > Dear Reviewer kGMU,
> >
> > As the deadline for the discussion period is approaching quickly, we would like to kindly remind the reviewer that we are waiting for your response. We are happy to answer any further questions or concerns you may have
> >
> > Best, Authors

---

### Official Review · Reviewer_oxzH · 2025-07-02

**Clarity:** 3
**Significance:** 2
**Originality:** 2
**Rating:** 4
**Confidence:** 5

**Summary:**

The paper proposes introducing high-pass filters (edge filters) as a single filter in neural networks to improve their performance. The core hypothesis is that deep features can be decomposed into semantic- and domain-specific information, where the semantic information resides in the high-frequency components of the features and the domain information resides in the low-frequency components of the features. Edge filters operate by subtracting a mean-filtered features from the features, creating essentially a high-pass filter.

**Questions:**

I would appreciate a response to the weaknesses listed above, and would consider raising the score if those are addressed appropriately.

**Ethical Concerns:**

["NO or VERY MINOR ethics concerns only"]

**Final Justification:**

The authors provided additional experiments and addressed most of my concerns.
Hence I'm raising my score.

**Limitations:**

The authors provide a section listing the limitations. However, as mentioned in my listed weaknesses, I do not fully agree that those could not be (at least partially) resolved.

**Paper Formatting Concerns:**

No concerns

**Quality:**

2

**Strengths And Weaknesses:**

Strengths:
1. The method is simplistic and easy to integrate into existing architectures. The proposed method is architecture-agnostic and, therefore, is very compelling.
2. The paper provides empirical evidence that shows an improvement when using their method.

Weaknesses:
1. The clean cut between high-frequency semantics and low-frequency domain bias is an oversimplification. Other works show that high-frequency components in the input are a domain bias (Geirhos et al., 2018) and that utilizing more low-frequency responses in neural networks can increase their performance and robustness (Finder et al., 2024).
2. The linear probing observation in Section 3.1 is very similar (albeit still different in implementation) to an experiment by Wang et al. (2020), in which ResNet shows to be responding more to high frequencies in the input rather than low frequencies. Hence, it is not surprising that the results of the experiment here favor the high-frequency filtering. A more thorough analysis of previous literature is needed there.
3. The support for the claims relies on previous research in domain adaptation. The authors interpret this as evidence for their claim, suggesting that aligning the first and second-order statistics supports their argument. While plausible, this is not proof, and the success of aligning BN statistics can have other explanations.
4. The sparse coding analysis is flawed. Using a high-pass filter on the input does result in a sparse output $h$, but it does not guarantee that $\alpha$ is also sparse.
5. The point made at the end of Section 3, regarding the use of multiple edge-filters in a network causing substantial information loss, warrants further investigation.
6. The experiments are conducted on dated architectures (ResNet and Wide-ResNet are from 2016) and relatively small-scale datasets (CIFAR, GLUE, etc.). While the authors point that out in the limitations and point at computational resources, there can still be better options than what's provided. E.g., it is common to use subsets of ImageNet (of 50,100,200 classes), and using more modern mobilenets and/or convnext will not require much more computational resources.

References:
Geirhos et al. "ImageNet-trained CNNs are biased towards texture; increasing shape bias improves accuracy and robustness." ICLR 2018.
Finder et al. "Wavelet Convolutions for Large Receptive Fields." ECCV 2024.
Wang et al. "High-frequency Component Helps Explain the Generalization of Convolutional Neural Networks." CVPR 2020.

---

> ### Author Rebuttal · Authors · 2025-07-31
>
> We appreciate the insightful feedback from the reviewer commenting on our method’s simplicity, architecture-agnostic applicability, and extensive experiments.
>
> **W1 Conflict with previous works:** Your point is that there are previous studies showing that low-frequency response is helpful in the input domain, but our claim focuses on high-frequency features, so it seems like we are making conflicting claims. However, our paper deals with high-frequency features in the **feature domain** of deep layers, not the input domain. This is also why our paper is titled “Deep” Edge Filter. Since we are addressing a different data level than previous works (deep features vs. input data), the two claims do not conflict. I hope this explanation helps clarify any misunderstandings you may have had.
>
> **W2 Similar paper:** Thank you for your suggestion for the paper by Wang et al. (2020). However, we respectfully disagree with your opinion that our paper is similar to that paper. This is because, recommended paper is about 1. Focusing on the image data - which is input level 2. Enforcing low frequency. Our paper deals with latent feature (i.e., deep feature) - which is feature level, and we also enforce high frequency feature in latents. So the scope is totally different.
>
> **W3 Empirical Justificationfrom from DA:** There seems to be a misunderstanding. We do not attempt to prove this using DA(domain adaptation), but rather take a conceptual approach to the problem based on empirical experience. We demonstrate a common phenomenon in deep learning models, namely that general features tend to exist mainly at high frequencies. Proving this phenomenon is considered impossible due to the complexity of deep learning, and this paper also stops at empirical observation. In other words, this paragraph is not a proof but an interpretation of the first and second order statistic matching techniques widely known in DA from a frequency perspective.
>
> **W4 Sparse coding:** Although the arithmetic density of deep features may not decrease immediately after HPF, when transformed into the frequency domain, the density is reduced. This is because the information filtering process of HPF occurs in the frequency domain. This density reduction phenomenon in the frequency domain can be directly confirmed in Fig. 3c. Another approach is that even though the output density decreases in the layers after HPF (Fig. 2a, 2b), the generalizable ability is maintained, which means that the model has acquired sparse coding ability. Since the only change in the model is the addition of HPF, it can be logically concluded that HPF has sparse coding ability.
>
> **W5 Multiple edge filters:** We understand your question concerns why multiple edge filters cause substantial information loss. This occurs due to cumulative frequency degradation: each edge filter (HPF) removes low-frequency components, and cascading multiple filters compounds this effect. When filters are applied sequentially, each filter is applied to already filtered data, resulting in exponential (quadratic) loss of information. This is why applying multiple edge filters causes severe information loss.
>
> **W6 Larger dataset:** The reviewer requested that we conduct experiments with larger datasets and modern architectures. The request for larger datasets was also a common question from reviewer Zz5q, so we felt the need for additional experiments, and we appreciate the advice on subset classes. Following the comments, we conducted experiments using the ResNet18 model on ImageNet200, which randomly selected 200 classes. The Deep Edge Filter applied a mean HPF with layer 2 and kernel size 11.
>
> |             |        Source | Direct | NORM | TENT |
> |-------------|:---------------:|:--------:|:------:|:------:|
> | baseline    | **58.8**          | 2.0    | 23.9 | 21.6 |
> | Edge Filter | 55.5          | **2.1**    | **25.8** | **23.2** |
>
> The results of experiments on ImageNet are also consistent with those obtained using the existing CIFAR dataset. Models applying Edge Filter show lower performance on general validation datasets but higher performance on corrupted (domain-shifted) data, demonstrating that they learn more general features during training. This proves that our claim is not limited to low-resolution datasets.
>
> **W6 Modern architecture:** This paper already presents experimental results using ViT and BERT, which are modern architectures based on transformers, so it would be helpful to refer to those results.
>
> In addition, we conducted additional experiments using the ViT-L-14 OpenCLIP model trained on the Laion2B dataset during the rebuttal process. We designed the training-free zero-shot classification scenario experiment using prototype matching methodology that measures the distance between the latent vector of the image to be predicted and a single data point, i.e., prototype, representing each class. Standard benchmarks PACS and DomainNet are used, with one randomly selected image per class from the source domain as the class prototype. The zero-shot classification accuracy of the shifted target domain images using the prototype is examined. The performance of the baseline case, which uses the embedding vector as is, and the case where the Edge Filter is applied are shown in the table below.
>
> |             |                |      PACS      |      | \| |               |   DomainNet   |      |
> |:-----------:|:--------------:|:--------------:|:----:|----|:-------------:|:-------------:|:----:|
> |             | Photo > Sketch | Sketch > Photo |  avg | \| | Real > Sketch | Sketch > Real |  avg |
> |   baseline  |      59.1      |      54.3      | 56.7 | \| |      24.8     |      39.2     | 32.0 |
> | Edge Filter |  **63.2**    |  **60.1**   | **61.7** | \| |   **25.0**  |  **39.5**  | **32.3** |

---

> > ### Author Response · Authors · 2025-08-04
> >
> > Dear Reviewer oxzH,
> >
> > As the deadline for the discussion period is approaching quickly, we would like to kindly remind the reviewer that we are waiting for your response. We are happy to answer any further questions or concerns you may have
> >
> > Best, Authors

---

> > > ### Author Response · Authors · 2025-08-07
> > >
> > > We sincerely thank the reviewer for their insightful feedback. As the deadline approaches, we would kindly remind you of our submission and would be grateful for your timely review, as our availability for further discussion is limited. To facilitate your understanding, we provide a summary of our core responses below, with a particular emphasis on the new experiments conducted as per your suggestions.
> > >
> > > * **[W1, W2] Distinction from Previous Works:** We clarify that our work focuses on the **feature space** of deep networks, whereas the referenced papers commonly address the **input space**. Therefore, our claims regarding frequency characteristics are not in conflict but are fundamentally different.
> > >
> > > * **[W3] Empirical Justification:** Our claims are based on an **interpretation** of phenomena in Deep Learning, drawing concepts from Domain Adaptation (DA), rather than a formal proof.
> > >
> > > * **[W4] Sparse Coding:** Our proposed filter endows the model with a **sparse coding capability**, leading to more robust and generalizable features. Even if data appears dense in the arithmetic basis, applying HPF sparsifies it in the frequency domain due to the basis difference, meaning sparsity may only be apparent after this transformation.
> > >
> > > * **[W5] Usage of Multiple Edge Filters:** The substantial information loss from using multiple filters is due to **cumulative frequency degradation**, which causes an exponential loss of information.
> > >
> > > * **[W6] Validation on Large-Scale Datasets and Modern Architectures (Key Improvement):** Addressing a critical point from the reviews, **we have conducted extensive additional experiments** during the rebuttal period. These include tests on the **ImageNet dataset** and with a modern **ViT-L-14 architecture**. The results strongly and consistently support our claims, demonstrating the effectiveness of our method beyond smaller-scale datasets.

---

> > > > ### Comment · Reviewer_oxzH · 2025-08-08
> > > >
> > > > The authors provided additional experiments and addressed most of my concerns.
> > > > Hence I'm raising my score.

---

### Official Review · Reviewer_Zz5q · 2025-07-02

**Clarity:** 3
**Significance:** 1
**Originality:** 1
**Rating:** 5
**Confidence:** 3

**Summary:**

This paper introduces the deep edge filter, a technique that applies high-pass filtering to intermediate feature representations in deep neural networks to improve generalization. The authors hypothesize that neural networks encode task-relevant semantic information in high-frequency components while storing domain-specific biases in low-frequency components. Their edge filter, implemented as F_edge(h)=h-LPF(h), is applied to deep features across various modalities and architectures. The paper shows modest improvements on several benchmarks including test-time adaptation, sentiment analysis, NeRF rendering, and audio classification.

**Questions:**

- Dataset scale: Why were experiments not conducted on ImageNet or other large-scale datasets that are standard in the domain generalization literature?
- Comparison with SOTA: Can you provide comparisons with state-of-the-art domain generalization methods beyond basic baselines?

**Ethical Concerns:**

["NO or VERY MINOR ethics concerns only"]

**Final Justification:**

I change my mind. I believe this paper introduces well enough scientific information about generalization that is needed for the community.

**Limitations:**

Yes

**Quality:**

2

**Strengths And Weaknesses:**

Strengths:
- Universal applicability: The method works across different modalities and architectures
- Simple implementation: Requires only adding a single filter layer
- Improvements: Shows positive results

Weaknesses:
- Limited Novelty: The core idea of applying frequency filtering to deep features is not novel. As also cited by the paper, Lin et al. (2023) proposed "Deep Frequency Filtering for Domain Generalization" at CVPR 2023, which explicitly modulates frequency components in deep features using FFT during training.

- Insufficient Experimental Validation: The vision experiments rely primarily on CIFAR-10/100, which are considered toy datasets by modern standards. ImageNet experiments are notably absent despite being the standard benchmark for vision tasks. No comparison with state-of-the-art domain generalization methods or the closely related DFF work.

- Technical Concerns: The choice of mean filtering as the default LPF seems arbitrary and under-justified with limited ablation studies on this choice.

---

> ### Author Rebuttal · Authors · 2025-07-31
>
> We appreciate the insightful feedback from the reviewer commenting on our method’s universal applicability, simple methodology, and performance improvements.
>
> **W1 Novelty of paper:** Transforming data into the frequency domain for analysis or simply optimizing in the frequency domain is not a new idea. In the field of audio, it has long been common practice to apply convolution to spectrograms in the frequency domain. The DFF[2] cited in the review appears to be a work that tunes the optimization techniques in the frequency domain revealed in previous work[1]. However, we are not simply trying to solve the problem in the frequency domain. Our primary goal is to offer a new interpretation of how deep learning models store information, revealing a fundamental connection to human prior knowledge through a frequency-domain analysis. The core of this paper is the verification that deep learning stores a significant amount of generalizable information in the high-frequency region when viewed in the frequency domain.
>
> What is counterintuitive about our work is that, unlike previous work where “improved performance is a given” due to the nature of optimization, we achieved performance improvement through a non-trainable HPF where **“improved performance is not a given”**. While previous work has focused on proposing “techniques” that utilize frequency analysis, we propose a completely new "intuition" regarding how deep learning models form semantic features.
>
> **W2&Q1 Larger dataset:** We agree with the reviewer's opinion that our methodology would be more convincing if validated on larger datasets such as ImageNet. Taking into account the advice of reviewer oxzH, we conducted image modality experiments using the ImageNet200 dataset, which consists of 200 randomly selected classes from ImageNet. The results of the experiment using the ResNet18 model on ImageNet200 are as follows. For Deep Edge Filter, we applied a mean HPF with layer 2 and kernel size 11.
>
> |             |        Source | Direct | NORM | TENT |
> |-------------|:---------------:|:--------:|:------:|:------:|
> | baseline    | **58.8**          | 2.0    | 23.9 | 21.6 |
> | Edge Filter | 55.5          | **2.1**    | **25.8** | **23.2** |
>
> The results of experiments on ImageNet are also consistent with those obtained using the existing CIFAR dataset. Models applying Edge Filter show lower performance on general validation datasets but higher performance on corrupted (domain-shifted) data, demonstrating that they learn more general features during training. This proves that our claim is not limited to low-resolution datasets.
>
> **W2&Q2 Comparison with SOTA:** We applied our proposed method to a semi-supervised DG task, one of the most challenging tasks in the DG field. As the baseline method, we used UPCSC[3], the SOTA method presented at CVPR 2025. We kept the model architecture and training procedure unchanged and conducted experiments using one labeled sample per class from the PACS dataset, which is the standard DG benchmark. We used ResNet18 as the model backbone and applied the same edge filter configuration as in the ImageNet experiment: a mean HPF with layer 2 and kernel size 11. The results of the SSDG experiment are as follows.
>
> |                       |   A  |   C  |   P  |   S  |  avg |
> |------------------|:----:|:----:|:----:|:----:|:----:|
> | UPCSC | 34.3 | 20.2 | 17.2 | 22.9 | 23.7 |
> | + Edge Filter | **41.6** | **43.9** | **21.1** | **47.9** | **38.6** |
>
> When the edge filter was applied, the classification accuracy improved significantly from an average of **23.7%->38.6%**. This shows that there is still plenty of room for performance improvement simply by applying the edge filter proposed in the SOTA model, and indicates that the edge filter effectively improves the generalization performance of the model.
>
> **W3 Choice of filter type:** The hypothesis presented in Section 3.1 applies generally to HPFs regardless of the type of filter. In fact, when comparing the results implemented using various filters such as mean, median, and Gaussian in the ablation study in Table 5, we observed a consistent trend in most cases. Among these, the mean filter was used as the main filter because it is simple to implement and computationally efficient, not because it has any particular advantages. Even if another type of LPF is used instead of the mean filter, the experimental trend will remain the same, and the claims made in the paper will not change. We appreciate your consideration of this point.
>
> **References:**
> [1] Li, Zongyi, et al. "Fourier Neural Operator for Parametric Partial Differential Equations." International Conference on Learning Representations. 2021.
> [2] Lin, Shiqi, et al. "Deep frequency filtering for domain generalization." Proceedings of the Computer Vision and Pattern Recognition Conference. 2023.
> [3] Lee, Dongkwan, et al. "Unlocking the Potential of Unlabeled Data in Semi-Supervised Domain Generalization." Proceedings of the Computer Vision and Pattern Recognition Conference. 2025.

---

> > ### Author Response · Authors · 2025-08-04
> >
> > Dear Reviewer Zz5q,
> >
> > As the deadline for the discussion period is approaching quickly, we would like to kindly remind the reviewer that we are waiting for your response. We are happy to answer any further questions or concerns you may have
> >
> > Best, Authors

---

> > ### Comment · Reviewer_Zz5q · 2025-08-06
> > **Core Concerns Unresolved, I decrease my point**
> >
> > I thank the authors for their response, I apologize greatly for the late response. I reviewed the paper once again, and I also looked at the related work for better judgment, specially Lin et al. I read the other reviews and I know two of them accepted the paper, but unfortunately I find your response insufficient to address the core concerns.
> >
> > ## Novelty:
> > - I tried to understand this part so much but I couldn't figure it out. The authors' distinction between their "non-trainable HPF" and DFF's trainable approach is unconvincing, distinction is minor. Their argument that they offer a new "intuition" rather than just a "technique" is changing the language without adding real substance. The fundamental concept (applying frequency filtering to deep features) remains the same as DFF.
> >    - Lin et al. (2023): Transforms intermediate deep features to frequency domain using FFT, modulates them with learned masks, then converts back
> >    - Submitted paper: Transforms intermediate deep features to frequency domain using FFT, applies filtering (subtraction-based), then converts back
> > - The authors claim they work on "high-frequency features in the feature domain" while Lin works on something different (to reviewer oxzH) but I can't see that. I believe both papers work on the frequency domain of deep layer features.
> > - While the specific operations differ (multiplication vs. subtraction), both papers share the same core insight, modulating frequency components of deep features improves domain generalization. Maybe I'm missing something but I can't understand the intuition this paper is adding, when your paper it not a technique paper, then it must discover scientific insights of high importance.
> >    - Lin et al.: Learns instance-adaptive attention masks that multiply with frequency representations.
> >    - Submitted paper: Uses fixed filtering operations to create high-pass filtering.
> > - Finally the authors claim: "we are not simply trying to solve the problem in the frequency domain. Our primary goal is to offer a new interpretation of how deep learning models store information", but I can't agree with that. Lin et al. is not just "tuning optimization techniques", but they propose a conceptual framework about how different frequency components affect domain generalization, very similar to what this paper claims. The intuition is already there.
> > - in other term, from what I can understand from these paper, both papers make similar conceptual claims:
> >    - Lin et al.: different frequency components are of different transferability across domains
> >    - This paper: deep learning stores generalizable information in high-frequency region
> >
> > ## Others:
> > - I believe the authors response for mean is weak. Your response that the mean filter was chosen for simplicity confirms my concern about arbitrary design choices. Your Table 5 shows median sometimes performs better, so if this paper is going to be intuitional, what is the justification for choosing mean? this should be explained mathematically as part of the hypothesis. I can't find theoretical or empirical justification for why mean is optimal. It look arbitrary.
> >
> > - ImageNet200 is not full image net [and in this small dataset the The improvements are marginal (2.0 to 2.1 NORM, 21.6 to 23.2 TENT)], This is far from the standard validation expected for a method claiming universal applicability. Being in this field for a while, I know how misleading small datasets can be. There are many experiments showing promising results at low scale but lose all their advantages on big datasets. I can't trust small datasets.
> >
> > - The experiment is fairly insufficient. The SOTA comparison is single-case (UPCSC + Edge Filter), not a head-to-head against the best DG methods from literature. The lack of broader comparison leaves open whether the improvement generalizes across multiple SOTA baselines.
> >
> > ## Final Judgment:
> > MY opinion can still change. I would try to engage with other reviewers to see if I would change my mind. But for now, I don't think this paper is borderline, and I should change my score to reject. \
> > I thank the authors one again for putting the time to rebuttal, and I'm sorry if I misunderstood their paper, but this is my honest opinion at this time.

---

> > > ### Author Response · Authors · 2025-08-07
> > >
> > > Dear reviewer, thank you for taking the time to respond and for giving us the opportunity to discuss our paper. Above all, we are pleased that we have finally identified the misunderstanding between us and addressed it. We believe this is the key point where the reviewer encountered difficulties in understanding our paper. The reviewer stated the following regarding our paper:
> > > > Transforms intermediate deep features to the frequency domain using FFT, applies filtering (subtraction-based), then converts back
> > >
> > > **Our paper does not use FFT at all.** Our methodology does not involve an explicit transition to the frequency domain; instead, we add a filter after the activation layer of the standard model. We explained our method in Sec. 3, which may be helpful for reference. We only use a smoothing-like filter to extract low-frequency components. While Lin et al. aims to gain performance by training parameters in the frequency domain, our work seeks to unveil the generalization prior of existing deep learning models from a frequency perspective. Thus, the two works are significantly different.
> > >
> > > To avoid such misunderstandings, we will specify separately in the final version that we do not use FFT. Below, we will respond point-by-point to the comments provided by the reviewer.
> > >
> > > Below, we will respond point-by-point to the reviewer's summary.
> > > ## Novelty
> > > 1. Our method DOES NOT use FFT at all.
> > > 2. The most important difference between Ours and Lin et al.'s operation is whether the frequency filter is optimized or fixed. It is natural for the model's accuracy to increase when the deep learning layer is optimized, but it is not natural for a fixed filter to improve model performance. It is difficult to accept that the operation of a trained learnable layer and a fixed filter are similar. Please consider the conceptual difference between learnable parameters and fixed digits in deep learning.
> > > 3. We will quote the reviewer's response.
> > > Lin et al.: Different frequency components have different transferability across domains -> This paper suggests the “technical possibility” of minimizing domain shift using frequency components.
> > > This paper: Deep learning stores generalizable information in the high-frequency region -> This paper suggests an “interpretation” of how deep learning models store information.
> > > As I see it, the reviewer's response already highlights sufficient differences between the claims of the two papers. Lin et al. focus on proposing a “technique” to solve domain shift, while we focus on presenting an “interpretation” of how existing deep learning models store information.
> > > If our claim were a restatement of Lin et al.'s, our claim should be derivable solely from Lin et al.'s results. However, this is apparently not true. As the reviewer pointed out, since they optimize the frequency component, they cannot obtain interpretations like other deep learning models. Lin et al.'s results **cannot be used to derive our core argument**, “The model tends to store their generalizable feature in high frequency on deep feature domain.”
> > >
> > > ## Others
> > > 1. We have consistently stated that the mean filter is not optimal and is an arbitrary choice. The reviewer also seems to think that the mean filter is an arbitrary choice, so there is no need for us to argue about this point. The key point is that we did not deliberately choose a filter that is “favorable” to us. As the reviewer pointed out, the fact that the median filter sometimes performs better than the mean filter clearly shows that we chose the filter “arbitrarily.” We are claiming that our hypothesis holds true for all general LPFs. Since it has been mathematically proven that the mean, median, and Gaussian filters are all LPFs, no additional mathematical justification seems necessary.
> > > 2. ImageNet200 is a significantly larger dataset than the existing CIFAR10/100 experiments, and furthermore, it is the same as ImageNet in terms of scale. It just uses fewer classes at the same resolution. The reviewer seems to be concerned that we intentionally trained on a smaller dataset, namely ImageNet200. However, this was because we tried to address as wide a range of questions as possible within the limited time available during the rebuttal process. ImageNet200 was the experiment proposed by Reviewer oxzH, which was ImageNet (50,100,200), and we chose the largest subset from among them to address your concerns about large datasets at the same time. As Reviewer oxzH mentioned, **the ImageNet subset experiment is widely adopted.** Regarding the marginal improvement, we want to kindly notice that this stems from an oversight by confusing **Direct** with **NORM**. Actually, the performance improvement is significant in the TTA field, with NORM: 23.9>25.8 (+1.9) and TENT: 21.6>23.2 (+1.6). Our scope lies at **NORM and TENT**, the benchmark devised for measuring generalization ability.

---

> ### Comment · Reviewer_Zz5q · 2025-08-07
> **I change how I looked at this work**
>
> First, I want to thank the authors for writing a response and I apologize for my misunderstanding about the method. They are correct. I appreciate the patience they have and I'm looking forward for this discussion.
>
> ## I totally missed a point
> oh, I see! You used FFT only for the analysis. I'm sorry for not getting your method. I tried to skim the paper once again is this your method?
> 1. In the middle layers, apply a low pass filter (blur-like filter) to the feature maps.
> 2. Subtract the blurred version from the original (output = original - blurred) and name it edge filter. [btw, why do you do it like this? you can do "original - blurred" in just once kernel operation. For example if your filters are 1×3, and your blurred filter is (x,y,z), then "original - blurred" would be (-x, 1-y, -z) kernel, is there any reason you are not doing this or am I not getting something?]
> 3. We expect to see DG improvement.
>
> "But it is not natural for a fixed filter to improve model performance." I can see now what your point is if my previous points were right. You just hypothesize that removing low frequencies improve generalization.
>
> I just need one questions to get this completely. Please let me know if I'm correct.
>
> 1. Take a CNN in mind. Name previous feature map as F=(f1, f2, f3,...), and the next layers feature map would be G = (g1, g2, g3,...). We can right (g1 = f1*k1 + f2*k2 + ...). As far as I understood you take a blurring filter "b" and make a new feature map calling it edge filter, lets name it "eg1 = g1 - g1 * b = g1 * (diric - b)"[where Dirichlet function here is 1 in the middle and 0 else where), so we can simply write (g1 = f1*[k1*(diric - b)] + f2*[k2*(diric - b)] + ... ])\
> So we can actually write g1 as the same kernel operation as before, just replace k_i in  (g1 = f1*k1 + f2*k2 + ...) with [k_i*(diric - b)]. My questions about this are:
> - Am I not right? does your method let this?
> - If this is right, then this is a point in any model that the optimizer can converge. So shouldn't we expect that large datasets converge to this point in the space? doesn't ImageNet already converging to this point?
>
> BTW, now I understand better you are not suggesting any method, so even if the model converges to these kernels in Imagenet the intuition that you are giving is valuable. I'm willing to accept this paper enjoy it : )
>
> I give it a 4. In the mean time I'll read the paper once again without thinking about it's practical applications. It's possible that I increase it.

---

> > ### Author Response · Authors · 2025-08-08
> >
> > Thank you for understanding our response, and we are glad that we were able to clear up the misunderstanding. Here are our answers to the additional questions you asked.
> >
> > We appreciate the reviewer's insight in suggesting that a high-pass filter can be implemented with a single kernel. This is partly correct and partly incorrect. As the reviewer mentioned, it is possible to implement a mathematically identical HPF using a single kernel with a mean filter. However, we also used various LPF (blurring method), including which cannot be implemented simply with a kernel like the **median filter**, so it was more efficient to implement it in the current subtract method for the reusability of the entire code.
> >
> > Regarding the question
> >
> > > Take a CNN in mind. Name previous feature map as F=(f1, f2, f3,...), and the next layers feature map would be…
> >
> > As we understand it, what you are saying is, “Even if we apply HPF, since it is a special case of CNN, increasing the size of the train dataset will cause other CNN layers to learn in a way that counteracts the influence of HPF, ultimately converging to a same optimal point regardless of whether HPF is applied or not.” Furthermore, it seems that this is the reason why you are requesting the ImageNet experiment.
> >
> > We believe that the HPF layer acts closer to a **regularization or activation layer** rather than a feature extraction layer. Our methodology is to make the low-frequency region sparse through HPF, which will act as a kind of regularization or modification of the activation pattern. For example, the reviewer mentioned the combination of edge filter and Conv layer, but what if we consider the combination of edge filter and activation layer instead? In fact, the edge filter is attached directly after the activation layer, and if we consider these two together as a new activation layer, it can be seen as having an orthogonal effect on the feature extractor (Conv layer). Additionally, the role of the edge filter is to induce sparsity in deep features, which is very similar to sparsity regularization such as L1 loss. Whether viewed from the perspective of activation layer modification or regularization loss, it is interpreted that the final converged parameter point of the model will change.
> >
> > In the rebuttal process, we conducted additional experiments using the CLIP model and confirmed that applying the edge filter improves the generalizability of the CLIP model as well. This demonstrates the effectiveness of our methodology even with the CLIP model trained on a very large dataset, and we hope this result will address the reviewer's concerns.
> >
> > |             |      PACS      |                |      |   DomainNet   |               |      |
> > |:-----------:|:--------------:|:--------------:|:----:|:-------------:|:-------------:|:----:|
> > |             | Photo > Sketch | Sketch > Photo |  avg | Real > Sketch | Sketch > Real |  avg |
> > |   baseline  |      59.1      |      54.3      | 56.7 |      24.8     |      39.2     | 32.0 |
> > | Edge Filter |      63.2      |      60.1      | 61.7 |      25.0     |    39.5       | 32.3 |
> >
> > We designed a training-free zero-shot classification scenario experiment using prototype matching methodology that measures the distance between the latent vector of the image to be predicted and a single data point, i.e., prototype, representing each class. Standard DG benchmarks PACS and DomainNet are used, with one randomly selected image per class from the source domain as the class prototype. The zero-shot classification accuracy of the shifted target domain images using the prototype is examined.

---

### Official Review · Reviewer_2BDh · 2025-07-03

**Clarity:** 2
**Significance:** 3
**Originality:** 2
**Rating:** 5
**Confidence:** 3

**Summary:**

This paper introduces the Deep Edge Filter which is a simple high-pass filtering technique applied directly to deep features of neural networks rather than inputs. The aim was to isolate high-frequency semantic information while suppressing low-frequency domain-specific bias. Their filter is constructed by subtracting a low-pass filtered version of the activations from the original feature map. The paper shows that this filter can enhance generalization performance across a wide range of data modalities like vision, language, 3d scene synthesis, and audio classification.

**Questions:**

1.⁠ ⁠Can the authors provide interpretability visualizations of the filtered vs. unfiltered deep features? Additionally, for the NeRF experiments, could the authors report view or region-specific metrics in high-frequency areas?

2.⁠ ⁠How does the performance of the model change based on where, within the network, the filter is applied? How does the edge filter behave in deeper architectures or large models such as GPT-style Transformers?

3.⁠ ⁠Does the Deep Edge Filter complement other robustness strategies like dropout or batch norm?

**Ethical Concerns:**

["NO or VERY MINOR ethics concerns only"]

**Final Justification:**

The paper presents a straightforward but interesting approach worth investigating theoretically and experimentally (at large scales). Further, the authors clarified all questions and concerns with empirical evidence. Overall, with the new findings, I've raised my rating to an accept.  Authors are strongly encouraged to include all new findings in their final copy.

**Limitations:**

Yes

**Quality:**

2

**Strengths And Weaknesses:**

__Strengths__:

1.⁠ ⁠The main insight of the paper that semantic features lie in high-frequency components while domain-specific noise is in low-frequency components is well-motivated and generally applicable to many domains. Applying these techniques to internal representations is an interesting adaptation.

2.⁠ ⁠From the results (Table 2, 3, 4), it seems as though the Deep Edge Filter does well across diverse modalities like test-time adaptation (vision), sentiment classification (language), novel view synthesis (3D), and sound classification (audio). The paper also includes ablation studies to isolate and validate the effect of the edge filter.

3.⁠ ⁠The method proposed in the paper does not require any learnable parameters and can be applied as a single module in any existing architecture.

__Weaknesses__:

1.⁠ ⁠The paper would benefit from feature visualizations like PCA or spectrograms to understand the effect of their Deep Edge Filter.

2.⁠ ⁠While the NeRF results mention reduced floaters, this is unsubstantiated and would be better supported with patch-wise PSNR or SSIM metrics, especially around the regions of high-frequency/detail in the scene.

3.⁠ ⁠While it can be costly computationally to validate the methods on a wider variety of tasks, it is challenging to accept generalization claims when there are no results on large models like CLIP or SAM.

4.⁠ ⁠The paper lacks a theoretical framework, so a deeper mathematical analysis would elevate the work.

---

> ### Author Rebuttal · Authors · 2025-07-31
>
> We appreciate the insightful feedback from the reviewer commenting on our method’s general applicability, validations across multiple modalities, and clear ablation studies.
>
> **W1&Q1 Visualization of Deep Edge Filter:** Visualizing how the distribution of features changes when applying the Edge Filter, as suggested by the reviewer, would be very helpful in understanding the characteristics of the Edge Filter. Thank you for your valuable comment. Following your suggestion, we performed tSNE visualization in the visual domain. As a result, we confirmed that when the deep edge filter was applied, the clusters for each class were more clearly distinguished. We will add this figure to the Supplementary Material. Please understand that we are unable to attach the image directly due to image attachment restrictions in this rebuttal.
>
> **W2&Q1 Region-specific NeRF metric:** To see the quantitative result of floaters, we calculated the patch-wise SSIM score for those scenes containing floaters. As for PSNR, the score explodes when the pixel scores are exactly the same, so we selected SSIM. We divided 400x400 generated scene images into 4*4 (16 patches), then calculated the SSIM score for each patch. Even though the patches are coarsely divided, there was a significant score margin between Ours and the Baseline, with the area floater occurring. The patches containing floaters show a severe gap compared to ours. We report the result as follows:
>
> | Scene | GT vs Base SSIM | GT vs Ours SSIM | SSIM Margin |
> |:-----:|:---------------:|:---------------:|:-----------:|
> | Drums |      0.919      |      0.996      |    +0.077   |
> | Drums |      0.936      |      0.990      |    +0.054   |
> | Drums |      0.775      |      0.824      |    +0.049   |
> | Chair |      0.849      |      0.999      |    +0.150   |
> | Chair |      0.618      |      0.739      |    +0.121   |
> | Chair |      0.714      |      0.822      |    +0.108   |
>
> **W3&Q2 Large models:** We verified whether our methodology could be applied to foundational architectures and evaluated its performance using the aforementioned CLIP. Based on the ViT-L-14 OpenCLIP model trained on the Laion2B dataset, we focused on the training-free zero-shot classification scenario. For the classification method, we designed the intuitive experiment using prototype matching methodology that measures the distance between the latent vector of the image to be predicted and a single data point, i.e., prototype, representing each class.
>
> To verify the generalization ability, we used the standard benchmarks PACS and DomainNet datasets. Specifically, we randomly selected one image per class from the source domain as the class prototype and compared the zero-shot classification accuracy of the shifted target domain images using this prototype. We selected the Photo & Sketch domain for PACS and the Real & Sketch domain for DomainNet, which are considered to be severe domain shifts. The performance of the baseline case, which uses the embedding vector as is, and the case where the Edge Filter is applied are shown in the table below.
>
> |             |                |      PACS      |      | \| |               |   DomainNet   |      |
> |:-----------:|:--------------:|:--------------:|:----:|----|:-------------:|:-------------:|:----:|
> |             | Photo > Sketch | Sketch > Photo |  avg | \| | Real > Sketch | Sketch > Real |  avg |
> |   baseline  |      59.1      |      54.3      | 56.7 | \| |      24.8     |      39.2     | 32.0 |
> | Edge Filter |  **63.2**    |  **60.1**   | **61.7** | \| |   **25.0**  |  **39.5**  | **32.3** |
>
> When applying the Edge Filter, we observed an improvement in classification accuracy from an average of **56.6%->61.7%** in PACS and from an average of **32.0%->32.3%** in DomainNet. This suggests that even simple post-processing in latent space using the Deep Edge Filter can effectively improve the generalization performance of models in understanding data from domains different from the training data.
>
> **W4 Theoretical framework:** Our work focuses on discovering phenomena through experimentation. In the field of deep learning, research has tended to proceed with discoveries made through experimentation first, followed by the establishment of theories to explain those phenomena. Considering this trend, our discoveries may serve as the basis for future theoretical research. We are the first to discover the association between generalizability and the frequency domain of deep features. Although we have not yet established a theoretical framework, we believe that subsequent research may uncover such a framework.
>
> **Q3 Complement other robustness strategies:** Yes, Deep Edge Filter can be used simultaneously with dropout and BN. In fact, Deep Edge Filter has already been used in conjunction with dropout and BN in experiments using convolution-based networks (WRN in image modality, sound modality). Since each experiment confirmed that using Deep Edge Filter resulted in additional performance improvements, Deep Edge Filter can be viewed as a complementary methodology to existing robustness strategies.

---

> > ### Author Response · Authors · 2025-08-04
> >
> > Dear Reviewer 2BDh,
> >
> > As the deadline for the discussion period is approaching quickly, we would like to kindly remind the reviewer that we are waiting for your response. We are happy to answer any further questions or concerns you may have
> >
> > Best, Authors

---

> > ### Comment · Reviewer_2BDh · 2025-08-04
> >
> > Thanks for addressing the questions and concerns. I appreciate the author's candid response regarding theoretical explanations for the Deep Edge Filter.
> >
> > Regarding floaters, it's clear that the method provides a substantial advantage compared to base methods.
> >
> > Lastly (as mentioned in Q1), can the authors comment on whether using the Deep Edge filter improves convergence (either observed in convergence time or image quality, such as PSNR) on high-frequency features, alleviating spectral bias in coordinate networks?
> >
> > Given the above clarifications from the authors, I'll raise my score.

---

> > > ### Author Response · Authors · 2025-08-05
> > >
> > > The reviewer asked about the convergence of the NeRF model's high-frequency features when the Deep Edge Filter was attached (convergence time and image quality). To verify the convergence of our method, we conducted an additional evaluation using an early-stage model that underwent 10,000 iterations of training out of a total of 140,000 iterations. The results of inference using the early stage model showed that the base model not only had floaters but also aliasing, where the surface of the rendering target was torn, resulting in poor qualitative results. On the other hand, when the Deep Edge Filter was applied, there was no aliasing, and the number of floaters was significantly reduced. Please understand that we cannot show the results directly in images due to the rebuttal policy. Instead, we measured the patch-wise SSIM as suggested by the reviewer and presented the results in the table below. The patch-wise SSIM measurement method is the same as in the rebuttal. The measurement results show that even in the early-stage model, applying the Deep Edge Filter improves the metrics related to high-frequency features, indicating that the Deep Edge Filter enhances convergence on high-frequency features.
> > >
> > > | Scene | GT vs Base SSIM | GT vs Ours SSIM | SSIM Margin |
> > > |:-----:|:---------------:|:---------------:|:-----------:|
> > > | Drums |      0.874      |      0.969      |    +0.094    |
> > > | Drums |      0.914      |      0.963      |    +0.049    |
> > > | Drums |      0.600      |      0.662      |    +0.062    |
> > > | Chair |      0.914      |      0.969      |    +0.055    |
> > > | Chair |      0.432      |      0.557      |    +0.125    |
> > > | Chair |      0.370      |      0.396      |    +0.026    |

---

> > > > ### Comment · Reviewer_2BDh · 2025-08-05
> > > >
> > > > Thanks to the authors for their prompt responses. The authors have addressed my concerns regarding visualization and showing evaluation on large models.  The proposed method is straightforward but interesting and has potential for further theoretical and larger-scale experimental explorations. Also, based on the latest response, it seems that using the deep edge filter indicates faster convergence (especially for Drums) of NERF towards high-frequency features.
> > > >
> > > > I strongly encourage the authors to include all of the new findings in the final draft. I will raise my score to an accept.

---

### Note · Authors · 2025-08-12

First, we would like to thank AC for giving us the opportunity to recap our discussion session during the review process. Before discussing about the rebuttal, we will briefly review our paper. In a nutshell, our work is a paper which **hypothesizes that deep learning models store generalizable features in high-frequency regions of deep features and experimentally verifies this hypothesis across various modalities and architectures.** Our contributions can be summarized as follows.

- We introduce Edge Filters, which are consist of general high-pass filters (HPFs) directly applied to deep features, thereby promoting modality-agnostic extraction of generalizable features.
- Through experiments, we confirmed that Edge Filters are applicable to both CNN-based and ViT-based architectures and enhance the generalization ability of various modalities, including images, language, 3D, and audio.

During the review process, we received positive evaluations from all reviewers, who acknowledged the following strengths.

- Our discovery that the Edge Filter, a non-trainable layer, improves model performance is not trivial but rather a non-trivial and valuable discovery. (kGMU, Zz5q)
- The Edge Filter can be applied universally across architectures and modalities and can be easily added to existing models. (2BDh, oxzH, Zz5q, kGMU)
- Despite its simple methodology, it shows significant performance improvement in tasks that require generalizability across various modalities. (kGMU, Zz5q)

We conducted the following additional experiments during the rebuttal process and reported the results.

- We verified our hypothesis using an additional large model, CLIP (ViT-L/14).
- We verified our hypothesis using a larger-scale dataset (ImageNet200).
- We demonstrated that our methodology also works with the SOTA DG method (UPCSC, Lee et al., 2025).
- We quantitatively measured high-frequency noise in the 3D domain.

Additionally, we promise to clarify the following points in the revised version.

- We will clarify that our Edge Filter does not use FFT, as it may be confused with.
- As discussed with Reviewer 2BDh, we will include additional experimental results for the 3D domain NeRF experiments in the appendix.

We express our sincere gratitude to all reviewers and ACs who took the time to review our paper and provided insights on directions for improvement.

---

### Decision · Program_Chairs · 2025-09-17

**Decision:**

Accept (poster)

**Comment:**

This work examines the effect of high-pass filtering / edge filtering on deep representations and its resulting impact on predictions. The experiments cover multiple modalities—images, audio, text, implicit representation of 3D—and multiple architectures—convolutional networks, transformers, and NeRFs.

Four expert reviewers with expertise on deep learning, computer vision, signal processing, and multi-scale architecture agree on acceptance (2BDh: 5, Zz5q: 5, kGMU: 5, oxzH: 4). The area chair sides with acceptance, finding no reason to overrule, and agrees with the positive evaluation of the reviewers that the empirical results are likely to inform and attract further interest as an instance of designed filtering having a role for learned representations. As a further point for acceptance, work at the intersection of deep learning and signal processing is valuable as a bridge between topics, and so both communities can learn from and extend this work.

While the initial reviews were not as positive and in agreement, the authors provided a rebuttal, and substantial discussion followed.
Although the trajectories of the reviews, rebuttals, and discussion are long and span multiple threads, ultimately every reviewer converged on acceptance. The area chair has followed each thread and likewise finds that issues that would argue against acceptance have been resolved. Nevertheless, the authors are strongly encouraged to incorporate the detailed feedback in the reviews especially w.r.t. the clarity of the contributions and its relationship to prior work focused on frequency analysis of deep features and filters plus domain adaptation and domain generalization.

Miscellaneous feedback:

Note that this comment has no bearing on the decision for the paper and is purely offered as information for the authors. Including a discussion of broader related work, for instance the on the frequency analysis of deep features and filters, alias, and multi-scale processing, may attract further interest and reach a broader audience. Examples could include Context contrasted feature (CVPR 2018), Octave Convolution (ICCV 2019), FrequencyLowCut Pooling (ECCV 2022), Improving native CNN robustness with filter frequency regularization (TMLR 2023), and other works at the intersection of signal processing and deep learning.